# Combining gut microbiota modulation and chemotherapy by capecitabine-loaded prebiotic nanoparticle improves colorectal cancer therapy

Tianqun Lang[1,2,7], Runqi Zhu[1,3,7], Xiao Zhu[1,3], Wenlu Yan[1,3], Yu Li[4], Yihui Zhai[1,3], Ting Wu[5], Xin Huang[1,3], Qi Yin ®[1,2,3] ✉ & Yaping Li ®[1,2,3,4,6] ✉

Colorectal cancer (CRC) therapy efficiency can be influenced by the microbiota in the gastrointestinal tract. Compared with traditional intervention, prebiotics delivery into the gut is a more controllable method for gut microbiota modulatory therapy. Capecitabine (Cap), the first-line chemotherapeutic agent for CRC, lacks a carrier that can prolong its half-life. Here, we construct a Cap-loaded nanoparticle using the prebiotic xylan-stearic acid conjugate (SCXN). The oral administration of SCXN delays the drug clearance in the blood and increases the intra-tumoral Cap concentration in the CRC mouse model. SCXN also facilitates the probiotic proliferation and short chain fatty acid production. Compared with free Cap, SCXN enhances the anti-tumor immunity and increases the tumor inhibition rate from 5.29 to 71.78%. SCXN exhibits good biocompatibility and prolongs the median survival time of CRC mice from 14 to 33.5 d. This prebiotics-based nanoparticle provides a promising CRC treatment by combining gut microbiota modulation and chemotherapy.

Colorectal cancer (CRC), the third most diagnosed cancer worldwide, exhibits high morbidity and mortality, even in individuals younger than 50[1,2]. In addition to radiotherapy and chemotherapy that may exert adverse side effects on normal cells, assistant therapeutic tools that can elicit immune responses combined with chemotherapy exhibit promising results in treating CRC in preclinical studies[3,4]. In recent years, the gut microbiota has attracted increasing attention due to its roles in various diseases[5]. Several probiotics or their products can inhibit tumor development by suppressing inflammation and promoting anti-tumor immune responses[6,7]. Homeostasis of gut microbiota also affects the efficacy and side effects of common cancer treatments, such as chemotherapy, radiotherapy, and immunotherapy[8,9]. Therefore, regulation of gut microbiota is a promising approach to improving the efficacy of CRC therapy[10]. Fecal microbiota transplant (FMT) is the most direct approach for gut microbiota modulation in the treatment of several digestive diseases[11,12]. However, large-scale application of FMT is limited by the lack of management standards and risk of serious infection. Therefore, new strategies for the precise regulation of gut microbiota are required.

[1]State Key Laboratory of Drug Research & Center of Pharmaceutics, Shanghai Institute of Materia Medica, Chinese Academy of Sciences, Shanghai 201203, China. [2]Yantai Key Laboratory of Nanomedicine & Advanced Preparations, Yantai Institute of Materia Medica, Yantai 264000, China. [3]University of Chinese Academy of Sciences, Beijing 100049, China. [4]School of Chinese Materia Medica, Nanjing University of Chinese Medicine, Nanjing 210023, China. [5]Department of Pharmaceutics, School of Pharmacy, Nanjing Medical University, Nanjing 211116, China. [6]Shandong Laboratory of Yantai Drug Discovery, Bohai Rim Advanced Research Institute for Drug Discovery, Yantai 264117, China. [7]These authors contributed equally: Tianqun Lang, Runqi Zhu. ✉e-mail: qyin@simm.ac.cn; ypli@simm.ac.cn

Prebiotics are a group of substances that can be used by host bacteria to enhance the proportion of beneficial species to improve the anti-tumor immunity[13–15]. Xylan is a polysaccharide prebiotic that exhibits favorable water solubility, good biocompatibility, and probiotic-boosting capacity, suggesting its potential for biomedical applications[16,17]. Intriguingly, xylan remains intact in the upper gastrointestinal tract (GIT) and is only degraded by microbes in the lower GIT, making it suitable for intestine-targeting drug delivery[18]. However, to date, reports on the application of xylan for anti-cancer drug delivery have only investigated ordinary nanocarriers to increase the water dispersibility of hydrophobic drugs and use the enhanced permeability and retention effects of tumors[19,20]. The specific prebiotic role of xylan in treating cancer remains unknown. Therefore, a new delivery platform should be constructed to facilitate the application of xylan.

Currently, chemotherapy is the predominant anti-CRC therapeutic regimen[21]. However, systematically administered chemotherapeutic agents exhibit insufficient curative efficacy and severe side effects, indicating the necessity to develop rational alternatives[22]. Capecitabine (Cap), an inactive oral fluoropyrimidine carbamate, is metabolized via a three-step cascade after absorption, and converts into 5-fluorouracil predominantly in the tumor tissues[23]. Cap is the first-line treatment regimen for metastatic CRC[24]. Despite its tumor-specific toxicity and high response rate, the blood plasma half-life of Cap is too short (0.5–1 h), therefore, a high-dose administration is necessary twice daily, but it may cause dose-related toxicity[25,26]. Therefore, an effective delivery strategy that can delay its plasma clearance is essential to enhance the clinical performance of Cap.

In this study, we construct a Cap-delivering nanoparticle from a xylan derivative to simultaneously prolong the half-life of Cap, achieve its intestine-targeted delivery, and regulate the gut microbiota for CRC treatment (Fig. 1a). We conjugate stearic acid (Sa) with xylan and Cap to synthesize their amphiphilic derivatives, termed Sxy and Scap, respectively. Subsequently, the Scap-loaded Sxy nanoparticle (SCXN) in water is prepared. Then the effects of SCXN on tumor growth, gut microbiota, and anti-tumor immunity are analyzed in a CRC mouse model.

## Results

### Construction and characterization of SCXN

To enhance the drug loading efficiency of Cap in xylan, the hydroxyl groups on xylan and Cap were reacted with the carboxyl groups on Sa to form ester linkages and obtain amphiphilic derivatives of xylan and Cap (Supplementary Fig. 1a). Structures of Sxy and Scap were confirmed using ${}^1$H-nuclear magnetic resonance spectroscopy (NMR) (Supplementary Fig. 1b). The weight-average molecular weights (Mw) of Sxy and Scap measured by matrix-assisted laser desorption ionization time-of-flight mass spectrometry (MALDI-TOF MS) were 1299.5 and 904.3, respectively (Supplementary Fig. 1c). The molecular ratios of Sa:xylan and Sa:Cap in Sxy and Scap were calculated to be 1:1 and 2:1, respectively. Scap was completely hydrolyzed into Cap under the action of lipase in vitro and in the liver within 30 min after orally administered in mice (Supplementary Fig. 2).

Blank Sxy nanoparticle (BXN) and SCXN were prepared using the film dispersion method. BXN and SCXN investigated via transmission electron microscopy (TEM) appeared as spherical and homogeneous particles (Fig. 1b). The mean particle sizes of BXN and SCXN determined via dynamic light scattering (DLS) were 112.83 ± 1.26 and 152.37 ± 1.88 nm, respectively (Fig. 1c). The zeta potentials of BXN and SCXN were −20.40 ± 4.13 and −22.17 ± 0.69 mV, respectively (Fig. 1d). The negative charges were brought about by the 4-O-methyl-α-D-glucuronic acid units partially substituting C2 of the xylopyranose units in xylan[27]. This charge was beneficial to avoid the clearance by the monocyte-phagocyte system. The drug loading (DL) and

encapsulation efficiency (EE) of Cap in SCXN were 9.49 and 87.59%, respectively, according to the drug concentrations measured by high performance liquid chromatography (HPLC).

Drug release behaviors of Scap from SCXN in different media in vitro were investigated to predict the stability and environment-sensitivity of SCXN in vivo. In phosphate buffered saline (PBS) (pH 7.4), less than 7% of Scap was released from SCXN within 48 h (Fig. 1e). Particle size of SCXN decreased to approximately 120 nm after incubation with the artificial gastric juice (AGJ) for 30 min, but remained stable without any significant size change in the following 7.5 h (Fig. 1f). The reason for the particle size reduction may be that in AGJ with a low pH, the carboxyl groups in xylan were protonated and strong hydrogen bonds between the carboxyl groups and the hydroxyl groups were formed, which led to the shrinkage of the nanoparticle[28]. To mimic the dynamic and ecological features of the intestinal tract, PBS (pH 7.4) containing mouse caecal content (MCC) was selected as the medium[29]. In the MCC-containing medium, Scap was gradually released from SCXN, with an accumulative release rate of up to 75% at 48 h (Fig. 1e). As a contrast, the release of free Scap was rapid and complete (nearly 85%) in both media.

To further study the intestinal Cap absorption rate mediated by SCXN in vivo, free Cap and SCXN were orally administered in CT26 tumor-bearing mice. More than 60% of the free Cap was absorbed into the small intestine within 1 h, and more than 80% in 4 h (Fig. 1g). When delivered by SCXN, Cap showed a much slower absorption rate in the small intestine than the free drug, with only 53.1% detected at 4 h. The accumulative Cap release rates of both formulations reached approximately 90% at 24 h.

The uptake and transport of SCXN in the intestinal villi were detected using 1-pyrenebutyric acid-labeled Scap (Scap-PBA)-loaded Sxy nanoparticle (SCXN-PBA). Since objects entering the small intestine need to cross the mucus barrier first, only faint red fluorescence of free Scap-PBA and SCXN-PBA was detected on the surface of the villi at 0.5 h after gavage administration (Fig. 1h). At 1 h, the signals in both groups had spread inside the intestinal villi. Free Scap-PBA was distributed throughout the villi, while SCXN-PBA mainly aggregated around the villi surface. At 2 h, the fluorescence of free Scap-PBA became brighter, and SCXN-PBA started to diffuse. The transport of free Scap-PBA across the villi was nearly complete at 4 h, exhibiting concentrated fluorescence at the base and a negligible signal in the villi. For SCXN-PBA, the spreading in the villi continued, showing stronger fluorescence than that at 2 h.

### SCXN increases the intra-tumoral drug accumulation

To evaluate the influence of SCXN on drug accumulation in tumors, a luciferase-expressing CT26 (CT26-luc) tumor-bearing mouse model was built and orally administered with the infrared fluorescent dye 1,1'-dioctadecyl-3,3,3',3'-tetramethylindotricarbocyanine iodide (DiR)-loaded Sxy nanoparticle (Sxy-DiR) or as free DiR. As observed with the in vivo imaging system (IVIS), almost equal DiR signals appeared in the abdomens of the two groups at 1 h post administration (Fig. 2a). However, over time, the DiR signal intensity of the free DiR group weakened and could hardly be detected in the mice after 24 h. In contrast, the high signal intensity lasted for 24 h in the Sxy-DiR group. More importantly, there were obvious overlaps between the signals of Sxy-DiR and tumors at 8 and 24 h, indicating that the Sxy nanoparticle might mediate a continuous accumulation of drug in the tumor. To more directly observe the biodistribution of the free drug and the Sxy nanoparticle-loaded drug in the main organs, the IVIS images of ex vivo tissues were taken. Free DiR mainly accumulated in livers and tumors at the beginning, but rarely in tumors from 8 h onwards (Fig. 2b). Compared with free DiR, the distribution of Sxy-DiR in tumors was obviously higher after 8 h, maintaining a strong fluorescence intensity at 24 h. Moreover, Sxy-DiR presented a longer retention in the intestine. The signals of Sxy-DiR were stronger than those of free DiR in the

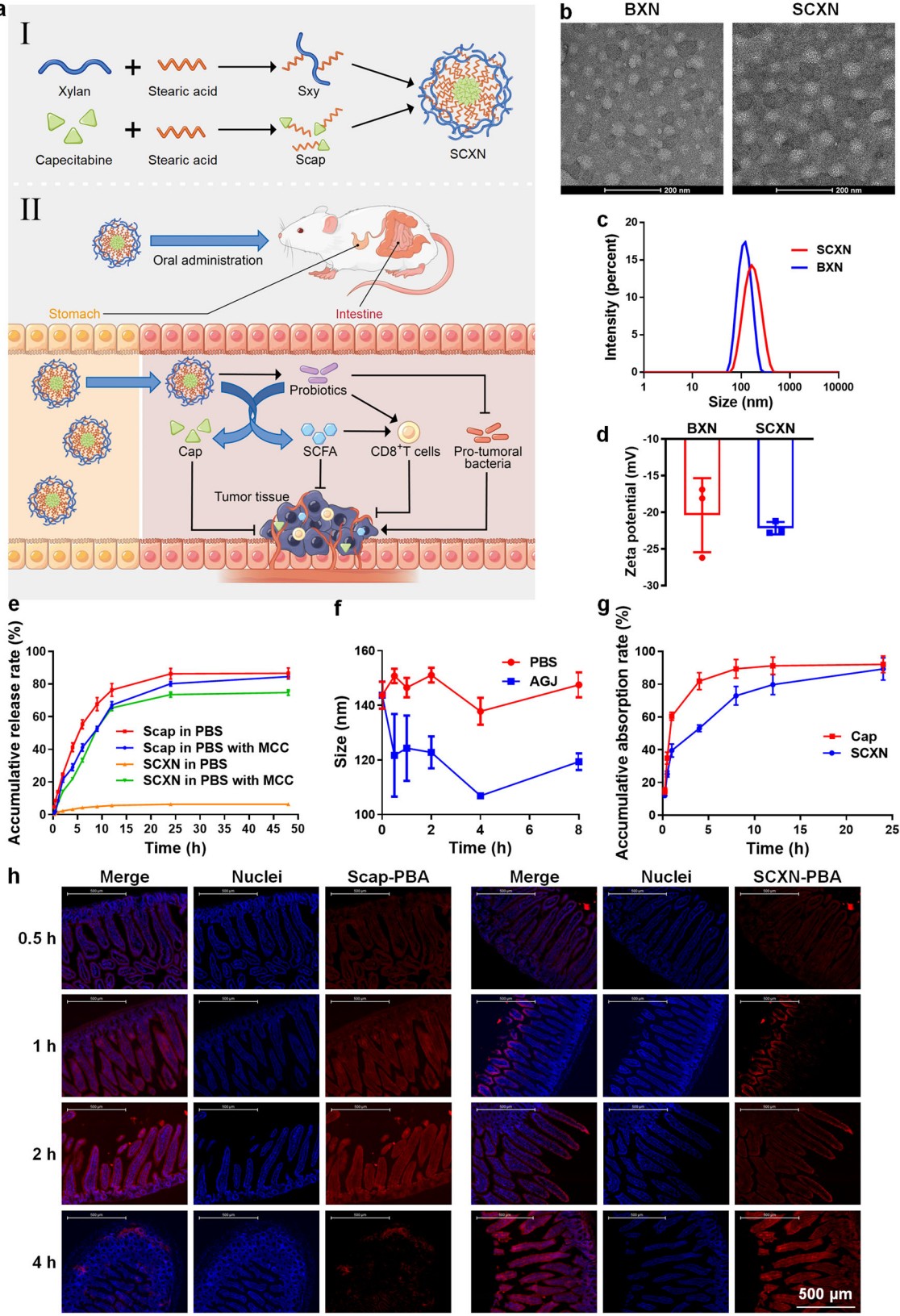

small intestine within 4 h and in the large intestine, where CT26 tumors were located, from 8 to 24 h (Fig. 2c).

Quantitative analysis revealed that the Cap concentrations in the livers, spleens and kidneys of the free Cap group were higher than those of the SCXN group within the first 4 h (Fig. 2d). However, higher distribution in these organs appeared at 8 h post-

administration with SCXN. The concentrations of 5-Fu, the active metabolite of Cap, in tumors of the SCXN group were kept at a relatively high level at 8–24 h, while those of the free Cap group decreased rapidly over time (Fig. 2e).

Pharmacokinetic profiles of Cap in the free Cap solution and SCXN were investigated in mice. Compared with the free Cap group,

**Fig. 1 | Construction and characterization of the Scap-loaded Sxy nanoparticle (SCXN). a** Schematic illustration of combining chemotherapy and gut microbiota regulation for CRC treatment by SCXN. **b** Transmission electron microscopy images of the blank Sxy nanoparticle (BXN) and SCXN. Scale bars: 200 nm. For each group, 3 independent samples were tested, and 1 representative image is shown. **c** Size distribution of BXN and SCXN. **d** Zeta potentials of BXN and SCXN. **e** Drug release curves of free Scap and SCXN in phosphate buffered saline (PBS; pH 7.4) with or without mouse caecal content (MCC). **f** Variation of particle sizes of SCXN in PBS at pH 7.4 and artificial gastric juice (AGJ) from 0 to 8 h. **g** Cap absorption curves in the small intestine after the oral administration of free Cap or SCXN in mice. **h** Distribution of the 1-pyrenebutyric acid (PBA)-labelled Scap (Scap-PBA)-loaded Sxy nanoparticle (SCXN-PBA) and free Scap-PBA along the villi of jejunum at different time points after oral administration. Scale bar: 500 µm. Data represent the mean ± SD. For each group, *n* = 3 independent samples in (**c**–**g**), and *n* = 3 mice in **h**. Source data are provided as a Source Data file.

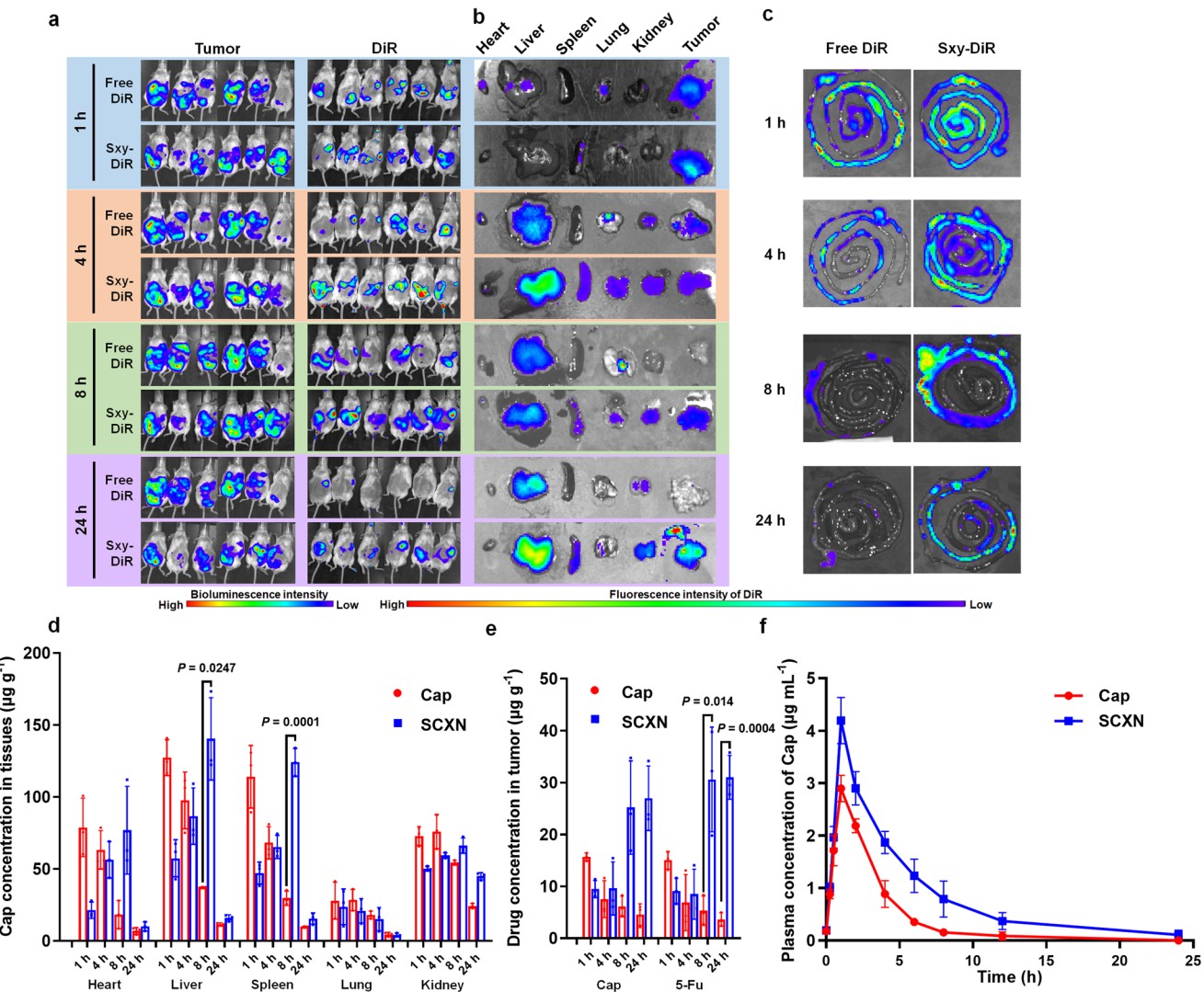

**Fig. 2 | Biodistribution and pharmacokinetics of SCXN. a** In vivo bioluminescence images of tumors (left) and the fluorescence of 1,1′-dioctadecyl-3,3,3′,3′-tetramethylindotricarbocyanine iodide (DiR) (right) in CT26-luc tumor-bearing mice at 1, 4, 8, and 24 h after the oral administrated of free DiR or DiR-loaded Sxy nanoparticle (Sxy-DiR). Six biological replicates are shown. **b** Ex vivo fluorescent images of tissues from CT26 tumor-bearing mice at 1, 4, 8, and 24 h after orally administrated with free DiR or Sxy-DiR. **c** Ex vivo fluorescent images of intestines from CT26 tumor-bearing mice at 1, 4, 8, and 24 h after the oral administration of free DiR or Sxy-DiR. **d** The content of Cap in different organs at 1, 4, 8, and 24 h after the oral administration of free Cap or SCXN in CT26 tumor-bearing mice. **e** Content of Cap and 5-Fu in tumors at 1, 4, 8, and 24 h after the oral administration of free Cap or SCXN in CT26 tumor-bearing mice. **f** Plasma concentration-time curves of Cap after the oral administration of free Cap or SCXN in mice. Data represent the mean ± SD (*n* = 3 mice). Statistical significance was calculated using unpaired two-sided *t*-test, with Welch's correction when variances are not equal. Source data are provided as a Source Data file.

SCXN showed an enlarged area under the concentration-time curve (AUC) of Cap, with increased half-life ($T_{1/2}$) and decreased plasma clearance (Fig. 2f, Supplementary Table 1). $AUC_{(0-\infty)}$ and $T_{1/2}$ of Cap of the SCXN group were 2.31 and 2.73 times of those of the free Cap group, respectively. Plasma clearance of the SCXN group was 43.18% of that of the free Cap group.

## Biocompatibility

The biocompatibility of a new drug delivery system is an essential issue that should be considered. During the safety evaluation, none of healthy mice receiving various formulations showed obvious body weight loss, and almost no pathological variation in the main organs was observed according to the hematoxylin and

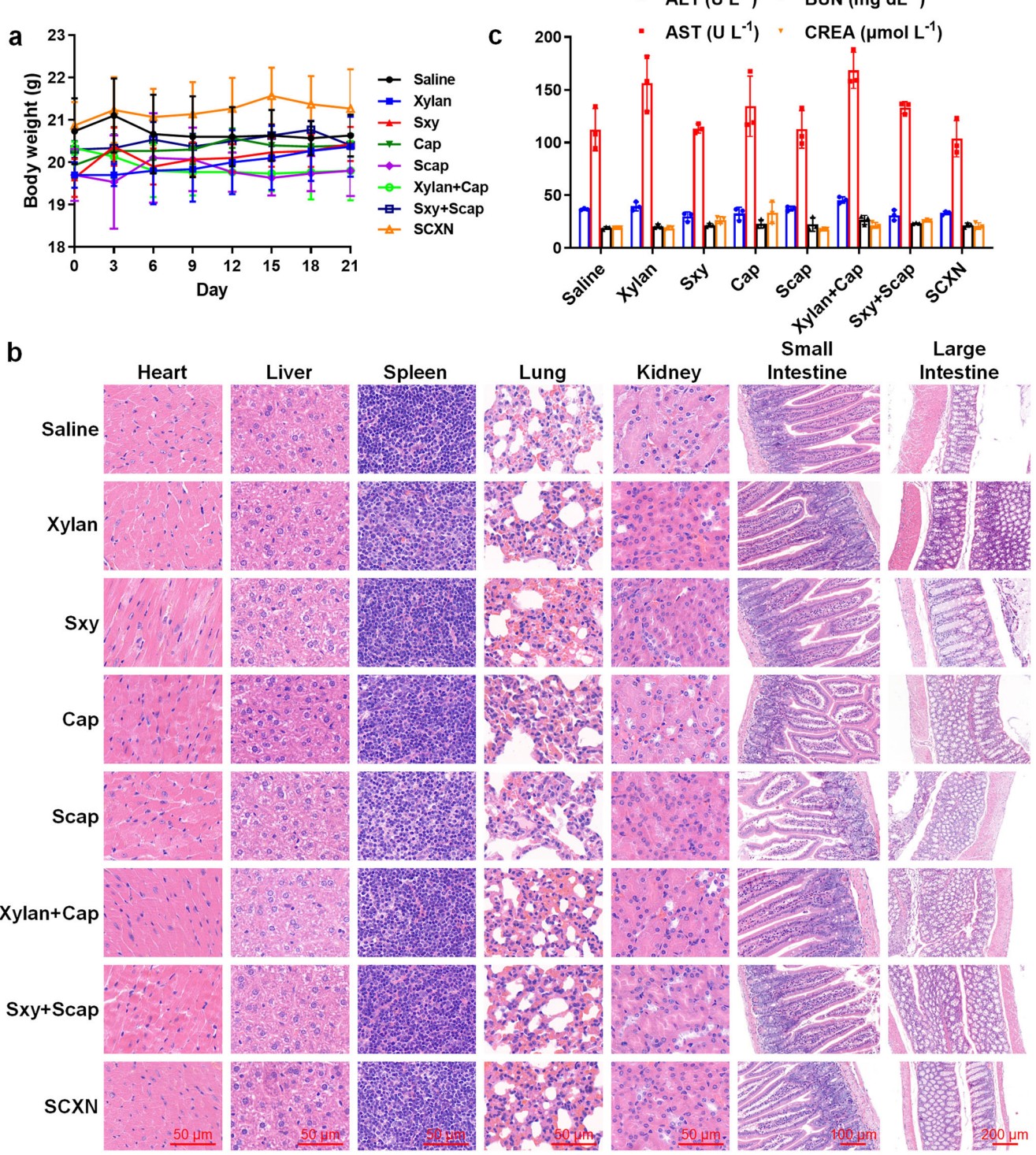

**Fig. 3 | Biocompatibility evaluation in healthy mice receiving multi-dose treatments with different formulations. a** Mice body weight variation during the treatment period. **b** Images of the hematoxylin and eosin (H&E)-stained sections of main organs. Scale bar: 50, 100, or 200 μm. **c** Biochemical parameters in blood. ALT alanine aminotransferase, AST aspartate aminotransferase, BUN blood urea nitrogen, CREA creatinine. Data represent the mean ± SD ($n = 3$ mice). Source data are provided as a Source Data file.

eosin (H&E)-stained sections (Fig. 3a–b). Notably, small intestines of the free Cap and Scap groups had shortened villi and crypts, indicating mucosal damage. When Cap was co-adminstrated with xylan, the length of villi and crypts remained normal. In addition, there was no difference in the blood biochemical index between the saline and SCXN groups (Fig. 3c). The red blood cell numbers in the blood samples of the Cap and Scap groups were slightly decreased (Supplementary Table 2), which was in accordance with a previous report[24].

## SCXN enhances the in vivo anti-tumor effects

To evaluate the anti-tumor effect of SCXN, tumor growth in CT26-luc tumor-bearing mice receiving different treatments was monitored by detecting the bioluminescence of tumor tissues with IVIS. The tumor

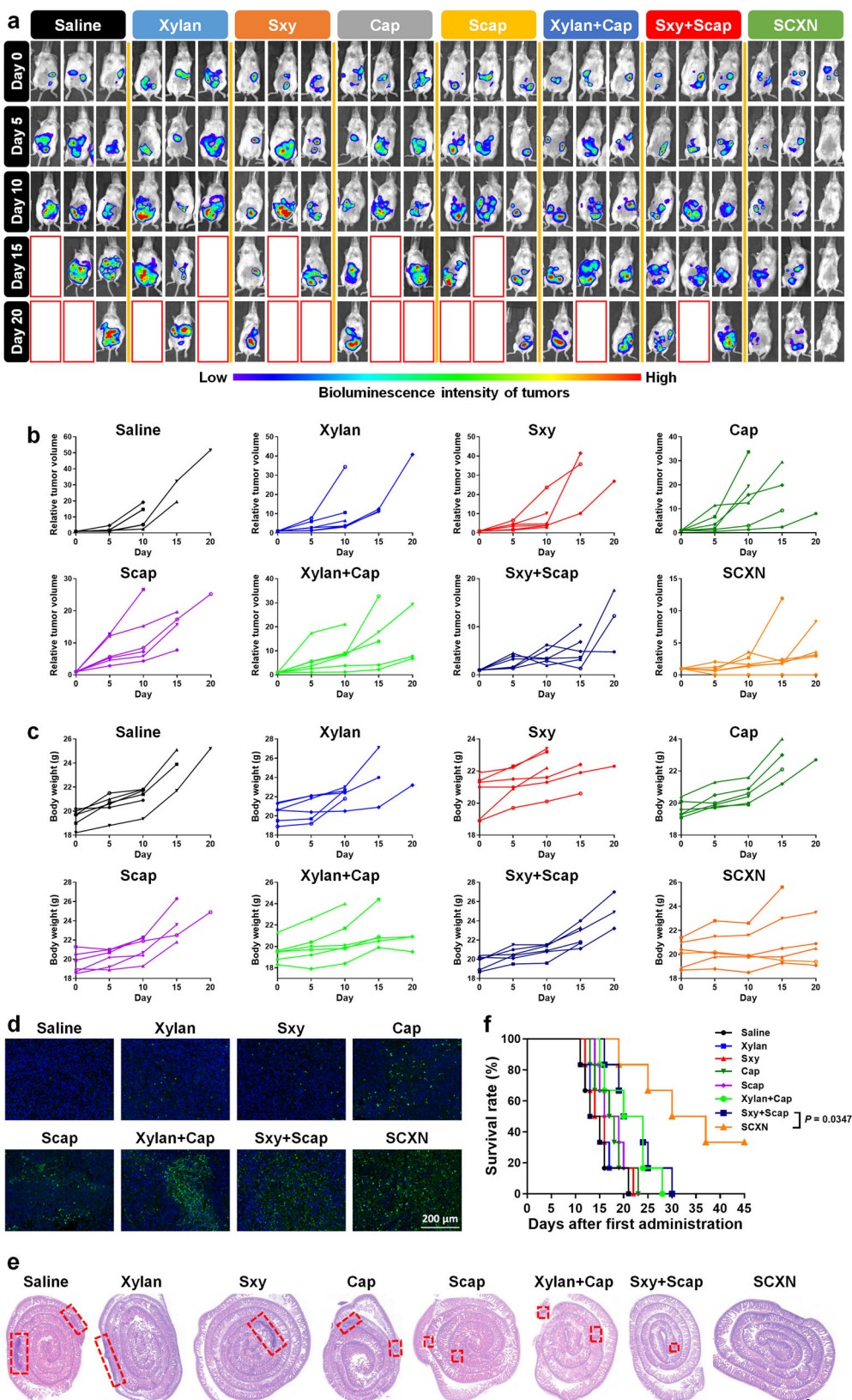

inhibition rates (TIR) of Cap and Scap alone were only 8.24 and 5.29%, respectively, which could be attributed to insufficient drug accumulation in tumors (Fig. 4a–b and Supplementary Figs. 3, 4). Although xylan and Sxy hardly inhibited tumor growth when used alone, their mixture with a chemotherapeutic agent displayed moderate tumor regression efficacy. Once mice were treated with co-delivered Sxy and

Scap in the nanoparticle form, the ascites decreased, and the bioluminescence emitted from tumor cells was maintained at a low level during the therapy period, reaching a TIR of 71.78%. In addition, the increase in body weights caused by ascites of the SCXN group was not as dramatic as that in the other groups, which revealed that SCXN could delay tumor progression (Fig. 4c and Supplementary Fig. 5).

**Fig. 4 | In vivo anti-tumor effects in CT26-luc tumor-bearing mice receiving multi-dose treatments of different formulations. a** In vivo bioluminescence imaging using an in vivo imaging system (IVIS). Three representative mice of each group are shown. Six biological replicates are conducted and shown in Supplementary Fig. 3. Red line boxes denote the dead mice. **b** Variation in relative tumor volumes calculated according to the amount of photons during the therapy period. **c** Variation in body weights during the therapy period. **d** Deoxynucleotidyl transferase-mediated dUTP-biotin nick end labeling (TUNEL) immunofluorescence examination of the CT26-luc tumor sections. Scale bar: 200 μm. **e** H&E-stained intestine sections from the CT26-luc tumor-bearing mice at the end of the therapy period. Red dot line boxes denote the tumor burdens. Scale bar: 2 cm. **f** Survival curves of mice within 45 d post the 1st administration. Data represent the mean ± SD. For each group, $n = 6$ mice in (**a**–**c**) and (**f**), and $n = 3$ mice in (**d** and **e**). A two-sided log-rank (Mantel–Cox) test was used for the statistical comparison of the survival study. Source data are provided as a Source Data file.

According to the deoxynucleotidyl transferase-mediated dUTP-biotin nick end labeling (TUNEL) immunofluorescence images of tumor sections, apoptosis was induced in most tumor cells in the SCXN group (Fig. 4d). Meanwhile, the tumor areas on the H&E-stained intestinal sections from mice treated with SCXN were hardly discernible compared with those of the control groups, in which there were multiple and large tumor burdens (Fig. 4e).

Prolonging patient survival is the ultimate goal of cancer therapy. A large proportion of mice treated with saline, xylan, and Sxy died during the therapy stage, with a median survival time of only 14–15 d. In the off-therapy stage, the lifespan of the SCXN group was significantly elongated and the median survival time reached 33.5 d (Fig. 4f). In contrast, the median survival time of the xylan+Cap and Sxy+Scap groups was only 22 d, with abdominal swelling before death.

Similar anti-tumor effects of different formulations were observed in another CRC model, the MC38 tumor mouse model. The tumor volumes were controlled at less than 500 mm³ by SCXN, while those in the saline and free Cap groups grew to 1547 mm³ and 1111 mm³ (Fig. 5a–c and Supplementary Fig. 6). SCXN achieved the a TIR of 87.33%, with a high rate of cell apoptosis in tumor sections examined by TUNEL (Fig. 5d). All body weights were relatively stable during the administration period (Supplementary Figs. 7, 8). This means that the anti-cancer ability of SCXN might be applicable to other CRC models.

## SCXN promotes the anti-tumor immune responses
Promoting anti-cancer immune responses is one of the essential roles that prebiotics play by modulating gut microbiota. The immune response elicited by SCXN was also investigated. First, the number of matured dendritic cells (DCs) per unit mass of draining lymph nodes of the SCXN group was 1.44 and 2.64 times higher than those of the xylan+Cap and Cap groups, respectively (Fig. 6a–b and Supplementary Fig. 9), indicating that SCXN enhanced DC maturation. In addition, compared with the Cap group, the intra-tumoral CD8⁺ T cells harvested from the SCXN group increased by 2.13 times (Fig. 6c–d and Supplementary Fig. 10). The results of the immunofluorescence assay were consistent with those of the flow cytometry detection (Fig. 6e). Prebiotics can relieve the immunosuppressive tumor microenvironment by suppressing cells that inactivate cytotoxic T lymphocytes. As one of the markers of the immunosuppressive tumor microenvironment, the proportions of regulatory T cells (CD4⁺Foxp3⁺ T cells, Tregs) in the tumor tissues of the saline and the Cap groups reached 134.2 and 98.16 per milligram of tumor, respectively (Fig. 6f–g). When mice were administrated with SCXN, the Tregs numbers in the tumors were significantly reduced, which was only 56.6% of that of the saline group. Furthermore, the CD8⁺ T cells to Tregs ratio of the SCXN group was 3.60 and 2.30 times of those of the Cap and xylan+Cap groups, respectively (Fig. 6h).

## SCXN enhances the proliferation of probiotics
The regulation of the gut microbiota by SCXN was confirmed by analyzing the fecal microbiota of mice receiving different treatments and at different stages of the chemotherapy course. According to 16 S rDNA identification, there was no significant change in community α-diversity among samples from mice receiving different treatments (Fig. 7a). Untreated and Cap-treated CT26 tumor-bearing mice displayed a distinct shift of community clustering in β-diversity non-metric multidimensional scaling (NMDS) analysis (Fig. 7b). At the class level, SCXN group exhibited the highest abundance of Clostridia (Supplementary Fig. 11). At the family level, the abundance of Lachnospiraceae and Ruminococcaceae was obviously elevated (Fig. 7c and Supplementary Fig. 12). At the genus level, the abundance of *Roseburia* and *Bifidobacterium* decreased in the saline group throughout the therapy period but increased in the SCXN group (Fig. 7d–e and Supplementary Figs. 13, 14). Loss of *Bifidobacterium* was slightly retarded by xylan alone. SCXN also induced a significant increase in the abundance of beneficial *Akkermansia* and *Faecalibaculum* and a decrease in the proportion of harmful *Desulfovibro*. An increase in xylan-digesting *Bacteroides* was observed after SCXN treatment or early administration of xylan. Notably, Cap or Sxy+Scap-treated mice exhibited a continuous increase in *Turicibacter*, *Rikenella*, and *Odoribacter*, while SCXN hardly induced a similar change. Cap-induced depletion of *Lactobacillus*, *Bacteroides*, and *Bacillus* was not observed in SCXN-treated mice. Although the relative abundance of the probiotics *Bifidobacterium* and *Blautia* increased at the middle therapy stage with Cap, it decreased to a comparatively low level at the end of treatment. The results were further confirmed by linear discriminant analysis effect size (LEfSe) analysis (Fig. 7f).

The modulation by prebiotics on the activity of probiotics is supposed to promote the production of short chain fatty acids (SCFA) in colons, which have been proven to inhibit CRC development through multiple mechanisms[30,31]. The influence of SCXN on SCFA production was evaluated by measuring the SCFA content using gas chromatography-mass spectrometry (GC-MS). Compared with the other four groups, the SCFA content in the feces of mice treated with SCXN was increased by 37% at least (Fig. 7g and Supplementary Fig. 15). The concentration of butyric acid, one of the most concerned beneficial SCFA for stimulating CD8⁺ T cells, in the fecal samples of the SCXN group was 3.53 and 3.27 times of those in the saline and Cap groups, respectively, which contributed to the enhanced anti-tumor immunity. Valeric acid and caproic acid concentrations were also elevated by SCXN.

## Discussion
The gut microbiome has been considered another "organ" of the host. Although their location is limited to the intestine, gut microbiota can maintain immune homeostasis of the whole body, thus influencing the progression and therapy of various types of cancer, not only CRC. A gut microbiota-regulating drug delivery system, SCXN, was constructed in this study to boost CRC therapy. SCXN was quite stable in the upper GIT, and the Scap release rate might be dependent on the degree to which Sxy was decomposed by the microbiota in the intestine. Owing to the controlled release behavior, the Sxy nanoparticles mediated a prolonged adhesion-uptake-transport process of drugs through the small intestine villi and delayed the absorption process of Cap.

According to the biodistribution and pharmacokinetics evaluation results, the free drug was quickly eliminated from the body, while the systemic distribution and metabolic process of SCXN in the blood slowed down, which demonstrated that the controlled release of Scap from the Sxy nanoparticle prevented premature drug clearance. SCXN ensured a high Cap concentration in the blood for a longer time, thus promoting the drug accumulation in tumors. Longer intra-tumoral

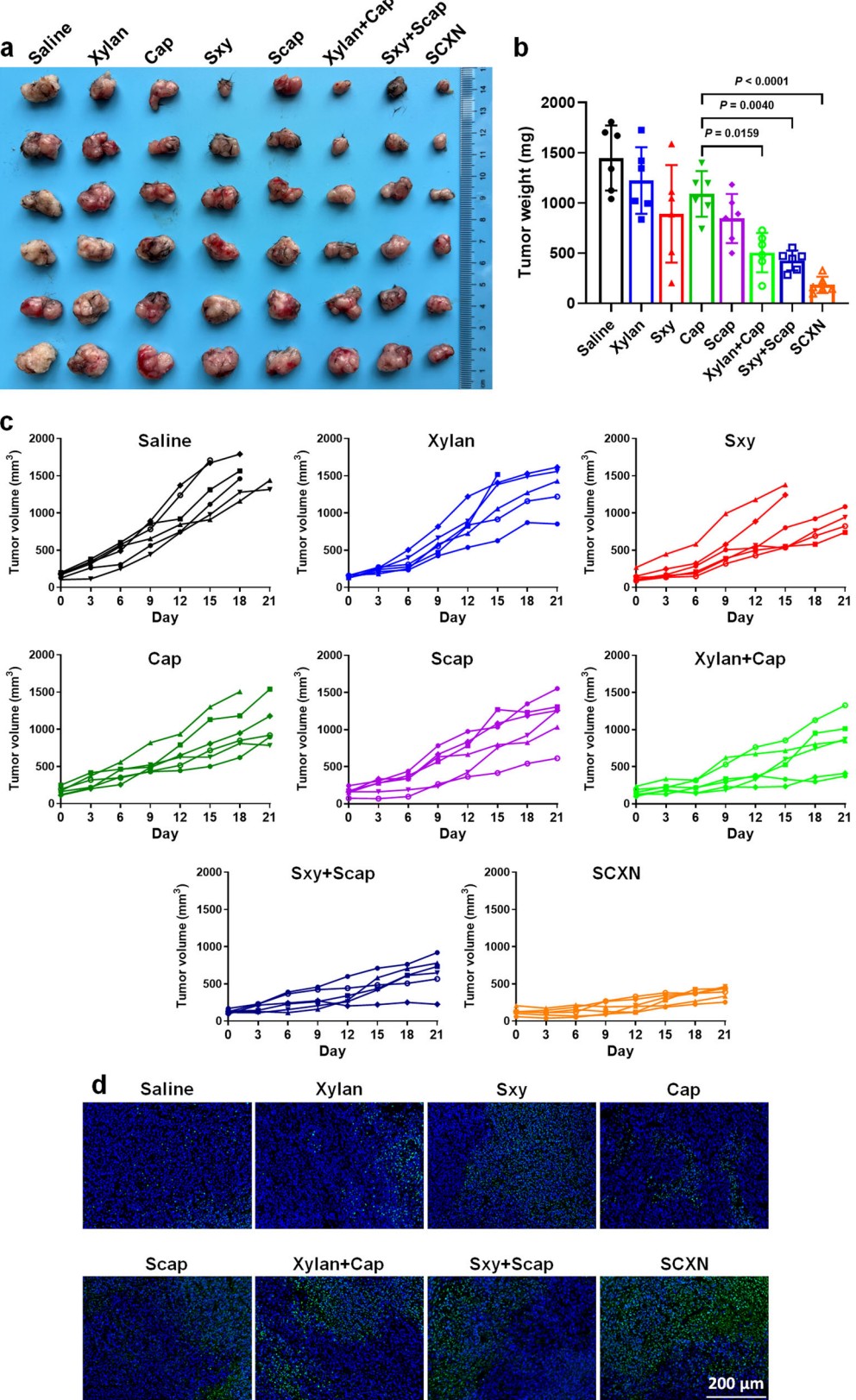

**Fig. 5 | In vivo anti-tumor effects in MC38 tumor-bearing mice receiving multi-dose treatments of different formulations. a**, **b** Tumor images (**a**) and average tumor weights (**b**) on day 21. **c** Variation in tumor volumes during the therapy period. **d** TUNEL immunofluorescence examination of MC38 tumor sections. Scale bar: 200 μm. Data represent the mean ± SD. For each group, $n = 6$ mice in (**a**–**c**), and $n = 3$ mice in (**d**). Statistical significance was calculated using one-way ANOVA with Tukey's multiple comparisons test. Source data are provided as a Source Data file.

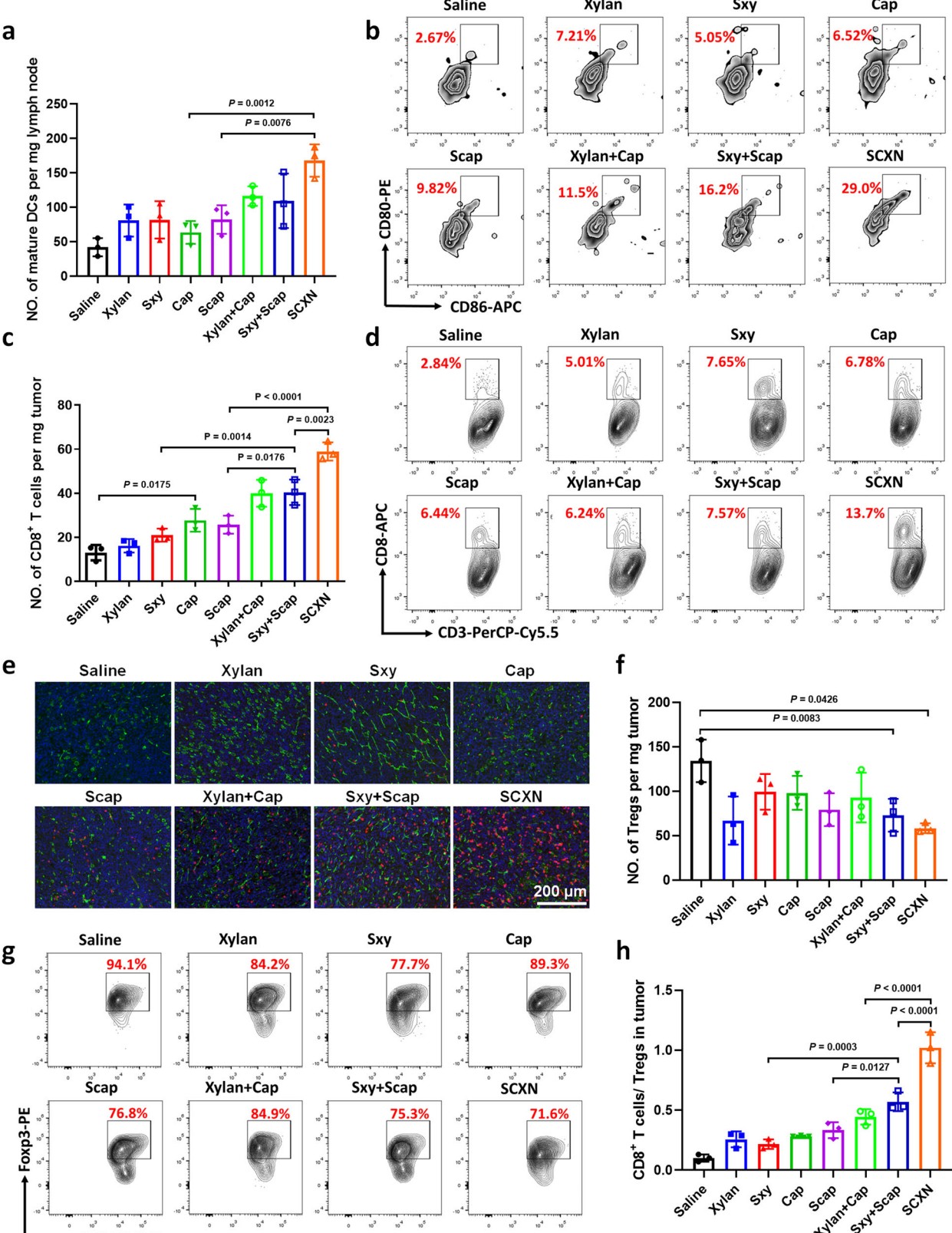

**Fig. 6 | SCXN promotes the anti-tumor immune responses.** Analysis of the numbers of immune cells in CT-26 tumor-bearing mice treated with multi-doses of different formulations via flow cytometry or immunofluorescence assay. **a**, **b** Numbers (**a**) and percentages (**b**) of mature dendritic cells (DCs; $CD80^+CD86^+$ cells gated on $CD11c^+$ cells) in draining lymph nodes. **c** Numbers of the $CD8^+$ T cells per mg of tumor. **d** Percentage of $CD8^+$ T cells in the total $CD3^+$ cell population in tumors. **e** Immunofluorescence images of tumor sections to examine the $CD8^+$

T cells (red fluorescence) infiltration. Green fluorescence: CD31. Scale bar: 200 μm. **f**, **g** Numbers (**f**) and percentage (**g**) of regulatory T cells (Tregs, $CD4^+Foxp3^+$ cells gated on $CD3^+CD4^+$ cells) in tumors. **h** Ratio of $CD8^+$ T cells to Tregs in tumors. Data represent the mean ± SD ($n = 3$ mice). Statistical significance was calculated using one-way ANOVA with Tukey's multiple comparisons test. Source data are provided as a Source Data file.

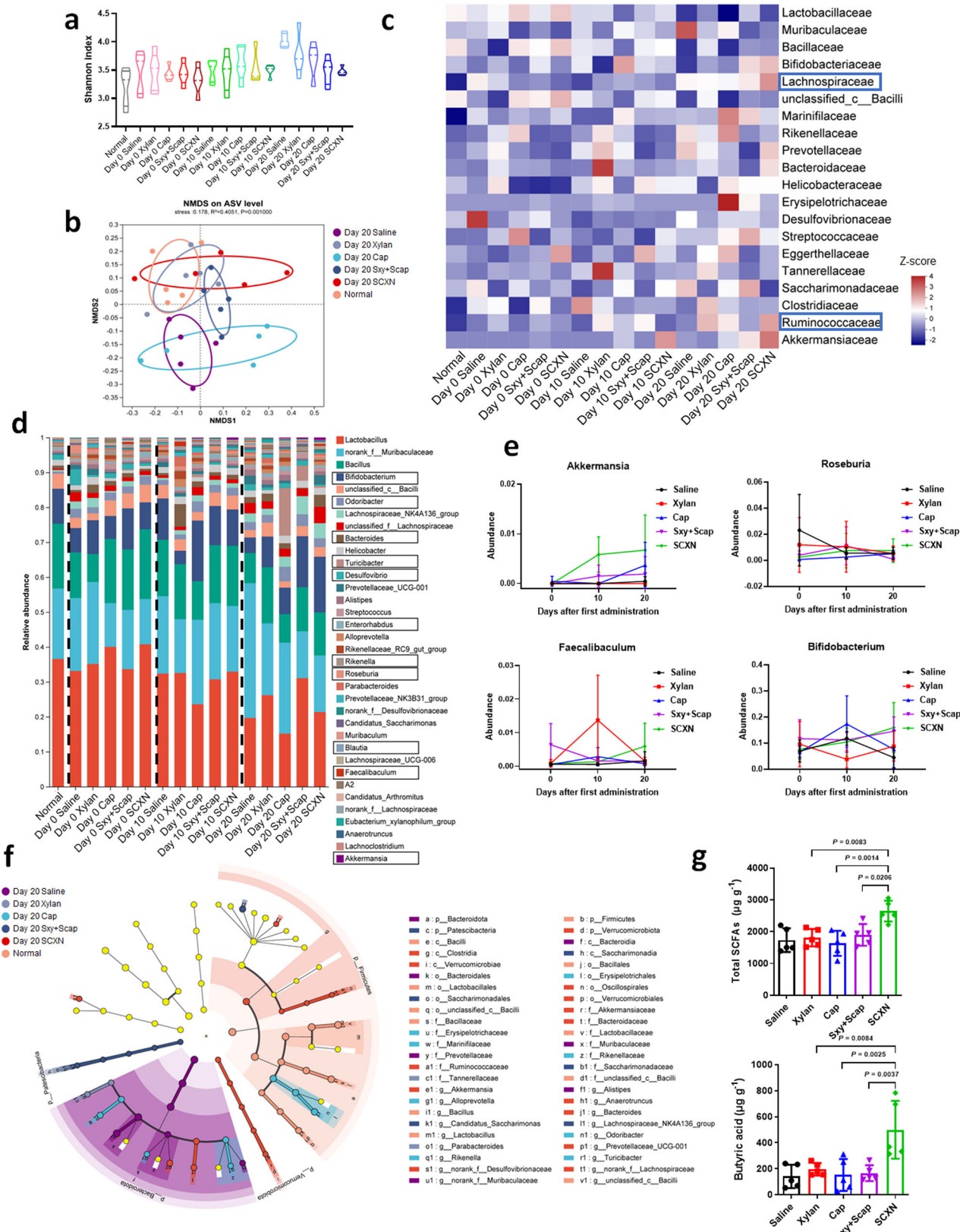

drug retention provides the opportunity to reduce the frequency of administration of Cap-based chemotherapy.

The biosafety evaluation results preliminarily confirmed that free Cap induced modest anemia, whereas SCXN exhibited good biocompatibility. The administration dose of Cap in this study (50 mg/kg per day for each mouse) was less than a tenth of that used in clinical applications (61 mg/kg per day for each human patient[32]). Therefore, the anti-tumor effect of free Cap was very limited. The encapsulation of Cap by SCXN exhibited efficient tumor-suppressing ability at the same low dose. SCXN relieved the immune escape and efficiently promoted the anti-tumor immune response. The experimental groups demonstrated a positive correlation between the increased probiotic

**Fig. 7 | Regulating gut microbiota by SCXN.** CT-26 tumor-bearing mice were treated with different formulations for 20 d and fecal samples were collected for 16 S rDNA sequencing before (Day 0), in the middle of (Day 10), and at the end of (Day 20) the treatment. **a** Microbial α-diversity in terms of Shannon index at the ASV level. **b** Microbial β-diversity NMDS analysis based on Bray-Curtis distance at the ASV level at the end of the treatment. **c** Heatmap showing relative abundance of the gut microbiota at the family level (displayed as normalized $Z$-score). **d** Barplot showing relative abundance of the gut microbiota at the genus level. **e** Change of

the relative abundance of *Akkermansia*, *Roseburia*, *Fecalibaculum*, and *Bifidobacterium* during 20-day treatment process. **f** LefSe analysis cladogram representing the significantly different taxas between different groups from phylum to genus levels at the end of the treatment (LDA > 2, $p$ < 0.05). **g** Summary of fecal SCFA levels and butyric levels in feces from CT26 tumor-bearing mice after different treatments. Data represent the mean ± SD ($n$ = 5 mice). Statistical significance was calculated using one-way ANOVA with Tukey's multiple comparisons test. Source data are provided as a Source Data file.

abundance and tumor suppression or anti-tumor immunity. Among the bacterial species influenced by SCXN, *Bifidobacterium* is a widely recognized probiotic that exerts protective functions in the digestive tract, and *Roseburia* exerts an anti-inflammatory effect in colitis pathology[33]. *Bifidobacterium*, *Akkermansia*, and *Roseburia* have been associated with improved tumor immunotherapy in clinic[34–37]. The anti-tumor effect of *Fecalibaculum* has been discovered[38,39]. *Desulfovibrio* is an endotoxin/$H_2$S-producing proteobacteria that is probably related to inflammation[40]. The blooming of *Bifidobacterium* caused by SCXN was associated with its prebiotic constituent, xylan, which has been widely known to promote *Bifidobacterium* growth[41,42]. Interestingly, the important xylan-degrading bacteria, Bacteroides, have been found to support the growth of other probiotics like *Roseburia* and *Bifidobacterium*, partly explaining the changes of these species in the SCXN-treated group[17,43,44]. Mice treated with SCXN displayed a different gut microflora composition from those receiving xylan only, probably due to the cooperation between xylan and Cap and the optimized delivery effect in vivo by this nano-system. This result suggests that the transformation of gut microbiota to a more anti-inflammatory and anti-carcinogenic state contributed to the promotion of anti-tumor immunity.

Chemotherapy usually induces intestinal toxicity and complicated changes in the gut microbiota. In this study, Cap used alone increased the levels of *Turicibacter*, *Rikenella*, *Enterorhabdus*, and *Odoribacter*, which are associated with intestinal inflammation and dysbiosis[45–49]. In addition, Cap reduced levels of certain beneficial species like *Bifidobacterium*. The negative impact probably resulted in the damage to the gut barrier and poor control of tumor progression at an advanced stage[50]. In contrast, SCXN achieved fine modulation without causing an increase of opportunistic pathogenic bacteria, and maintained *Bifidobacterium*.

SCFAs are the main products of anaerobic fermentation of indigestible polysaccharides and contribute to immunomodulation in inflammatory diseases and cancer. In this work, since mice in all groups were fed with the same food, which might be one source of SCFAs, the difference in gut SCFAs levels should be due to the distinct xylan intake. Compared to the acetate production pathway, which is common to most species, butyrate and propionate production is restricted to a much smaller group of bacterial types[51,52]. SCXN promoted the growth of various SCFAs producers, among which *Roseburia* (Lachnospiraceae) and Ruminococcaceae belong to the main butyrate-producing taxa, Clostridia, and *Bifidobacterium* and *Bacteroides* are the main acetate and propionate producers[52–54]. The efficient regulation of gut microbiota and enhanced SCFAs generation by SCXN built a tumor-suppressing and beneficial microorganism-supporting intestinal microenvironment. Consequently, the development of CRC was inhibited, and the lifespan of mice was prolonged. Xylan is a well-known prebiotic, whose health-promoting functions have been validated and linked with its effect of increasing the abundance or activity of probiotics[55]. Nevertheless, in this study, the effect of xylan alone on promoting anti-tumor immunity and facilitating SCFA production was limited. The reason might be that the SCFA-producing probiotics, which can degrade xylan, represented by *Lactobacillus and Bifidobacterium*[56], had a much lower abundance in the intestines of colon tumor-bearing mice than that in the normal ones. Xylan alone was not efficient enough to restore the balance of the intestinal flora.

Therefore, cooperation with tumor-killing agents or supplementation with exogenous probiotics is essential for the action of xylan in CRC therapy. The improved anti-tumor immune responses and tumor growth-suppressing efficacy of xylan+Cap compared with xylan and that of Sxy+Scap vs. Sxy indicated that the combination with Cap promoted the immune-regulating function of xylan. This result suggests that the mutual assistance between xylan and Cap can achieve a synergistic effect; thus, both are essential for treatment.

In conclusion, we demonstrated the potential of the combination of gut microbiota modulation and chemotherapy in CRC therapy via a single drug delivery system using prebiotics. Our results suggest that nanotechnology can aid in the development of more precise and controllable prebiotic-based delivery systems to treat different cancer types.

## Methods
This research complies with all relevant ethical regulations approved by the Institutional Animal Care and Use Committee (IACUC) of Shanghai Institute of Materia Medica, CAS.

### Materials
Xylan, stearic acid (Sa), 4-(dimethylamino)-pyridine (DMAP), anhydrous dimethyl sulphoxide (DMSO), dichloromethane ($CH_2Cl_2$), and N, N'-dicyclohexylcarbodiimide (DCC) were all provided by J&K Scientific Ltd. (Shanghai, China). Capecitabine (Cap) was obtained from Macklin Inc. (Shanghai, China). RPMI 1640 medium, fetal bovine serum (FBS), glucose solution, sodium pyruvate solution, trypsin-EDTA solution, and 4',6-diamidino-2-phenylindole (DAPI), fixation/permeabilization concentrate, permeabilization buffer, and fixation/perm diluent were all purchased from Thermo Fisher Scientific Inc (Waltham, USA). 1,1'-dioctadecyl-3,3,3',3'-tetramethylindotricarbocyanine iodide (DiR), D-luciferin potassium, penicillin G sodium solution, streptomycin sulfate solution and dialysis bag (MWCO 3.5 kDa) were purchased from Meilun Biotech Co., Ltd, (Dalian, China). All other reagents were purchased from Sinopharm Chemical Reagent Co. Ltd. (Shanghai, China) with analytical grade and used without further purification.

### Cell culture
CT26 murine colon carcinoma cells and CT26-luc cells were kindly provided by Prof. Yongzhuo Huang's lab in Shanghai Institute of Materia Medica, CAS. MC38 murine colon carcinoma cells (cat. no. FH0125) were obtained from Fuheng Biology Co. Ltd (Shanghai, China). All kinds of cells were cultured in RPMI 1640 medium which was supplemented with 10% FBS, 2.5 g $L^{-1}$ of glucose, 0.11 g $L^{-1}$ of sodium pyruvate, 100 U m$L^{-1}$ of penicillin G sodium, and 100 μg m$L^{-1}$ of streptomycin sulfate, and incubated with a humidified atmosphere containing 5% $CO_2$ at 37 °C.

### Animals
The female Balb/c mice aged 4−6 weeks (18−22 g) and female C57BL/6 mice aged 4−6 weeks (18−22 g) were obtained from Beijing Charles River Laboratory Animal Technology Co., Ltd (Beijing, China) and raised in the animal care facility in the SPF grade environment with sterilized food pellets and distilled water under a 12 h light/dark cycle. The normal chow included ≥18% protein, ≥4% fat, ≤5% crude fibre, ≤8%

ash, 1.0–1.8% Calcium and less than 10% moisture (Keao Xieli Feed Co., Ltd., cat. no. 1016706714625204224). The temperature for the housing room was ~24 °C and the humidity was ~50%. All animal studies were performed in accordance with the regulations approved by the Institutional Animal Care and Use Committee (IACUC) of Shanghai Institute of Materia Medica, CAS (Approval number: 2020-04-LYP-40).

### Establishment of animal tumor models
$1 \times 10^6$ CT26 cells or $5 \times 10^5$ CT26-luc cells were injected intraperitoneally into each Balb/c mouse to establish the CT26 or CT26-luc tumor mouse model. $1 \times 10^6$ MC38 cells were subcutaneously injected into each C57BL/6 mouse to establish the MC38 tumor mouse model.

### Synthesis of Sxy and Scap
Xylan (2.00 g), DCC (6.24 g), and DMAP (1.76 g) were dissolved in DMSO (80 mL). 4.31 g of Sa dissolved in DMSO (40 mL) was added dropwise, and then the mixture was stirred at room temperature (RT) for 24 h. Next, the resulting solution was added dropwise to 95% ethanol (2 L) under ice bath conditions, then the precipitate was collected by centrifugation (6000 g, 5 min) at 4 °C. Finally, the precipitate was resuspended in ethanol (30 mL) and dried in vacuum at 45 °C. Sxy was obtained as white oily solid.

Cap (359.35 mg), Sa (853.44 mg), DCC (680.89 mg), and DMAP (36.65 mg) were dissolved in dichloromethane (20 mL), stirring for 24 h at RT. Then the reaction mixture was filtered by filter paper, and the filtrate was washed and dried in vacuum. Scap was obtained as white powder.

Both compounds were strored at −20 °C until use. The structures of Sxy and Scap were confirmed via $^1$H NMR (AVANCE NEO 500, Bruker, USA), and the Mw were measured through MALDI-TOF MS (ThermoFisher Scientific, AB SCIEX 5800, USA).

### Hydrolysis of Scap in vitro and in vivo
4 mg Scap and 10 mg lipase were dissolved in 1 mL water. The mixture was incubated in 37 °C for 30 min, and 1 mL methanol was added. After vortex for 10 min, the mixture was centrifuged (2000 g, 2 min). The supernatant was collected and detected by HPLC (e2695, Waters, USA). Cap dissolved in methanol, Scap dissolved in methanol, and the mixture of Cap+Scap solution were detected by HPLC as controls.

Balb/c mice ($n = 9$) were orally administered with Scap (100 mg kg$^{-1}$ Cap). The livers of 3 mice were taken out at certain time points (0.5, 1, and 2 h), homogenized, and centrifuged (2000 g, 3 min). The supernatant was collected and detected by HPLC. The livers from mice without any treatment were taken out. 15.60 mg Cap, 300 mg liver tissues, and 1 mL methanol were mixed, homogenized, and centrifuged (2000 g, 3 min). The supernatant was collected and detected by HPLC as the control sample.

### Construction and characterization of SCXN
Sxy (366 mg) and Scap (140.4 mg) were dissolved in methanol (40 mL) in a round-bottom flask (500 mL). Then the methanol was removed through a rotary evaporator to form a film. Next, 18 mL of distilled water was added into the flask, and SCXN was formed under ultrasound. Finally, SCXN was filtered through 0.22 μm membrane. BXN without Scap was prepared with a similar method. The mean particle sizes and zeta potential of BXN and SCXN were measured by Zetasizer (Malvern, UK). The mean particle size variation of SCXN along with time from 0 to 8 h in PBS at pH 7.4 and AGJ (containing 16.4 mL 10% HCl and 10 g pepsase in 1 L AGJ) were tested, respectively. The morphology of BXN and SCXN was observed through TEM (Tecnai F20, FEI, USA). To determine DL and EE of Cap in SCXN, SCXN (1 mL) was mixed with methanol (9 mL). Then the mixture solution was vortexed and sonicated for 5 min, and centrifuged (3000 g, 10 min). The Cap concentration in the supernatant was determined by HPLC. DL and EE were calculated according to the following two equations:

$$DL\% = \frac{weight\ of\ the\ drug\ in\ the\ nanoparticle}{weight\ of\ the\ whole\ nanoparticle} \times 100\% \qquad (1)$$

$$EE\% = \frac{weight\ of\ th\ drug\ in\ the\ nanoparticle}{weight\ of\ the\ feeding\ drug} \times 100\% \qquad (2)$$

### Drug release in vitro
The release behavior of SCXN was analyzed by a dialysis bag diffusion method. The dialysis bags containing 1 mL of free Cap or SCXN (Cap concentration: 6.67 mg mL$^{-1}$) were placed in PBS (5 mL) with continued shaking at 37 °C. Then the medium was taken out at certain time points, and replaced by fresh medium. The concentration of Cap in the taken-out medium were measured by HPLC.

The influence of intestinal flora on drug release was studied in the MCC-containing medium. Caecal contents (2 g) were collected from the cecum of mice and dissolved in PBS (100 mL) immediately. After centrifuged (500 g, 5 min), the supernatant was used as the release medium. The drug release was analyzed by the same way as that in PBS. The taken-out medium was centrifuged (500 g, 5 min) again and purified through 0.45 μm membrane before tested by HPLC.

### Cap absorption in the small intestine
In order to study the release behavior of SCXN in vivo, CT26 tumor-bearing mice were randomly divided into two groups ($n = 21$ per group), the small intestines (4 cm) of 3 mice in each group were harvested at certain time points (0.25, 0.5, 1, 4, 8, 12, and 24 h) after the oral administration of free Cap and SCXN (50 mg kg$^{-1}$ Cap). Then the homogenate of the cut intestines and methanol were centrifuged, and the Cap concentration in the supernatant were measured by HPLC.

### Transport of SCXN in the small intestine
SCXN-PBA was prepared by the similar method as SCXN, replacing Scap with Scap-PBA. Balb/c mice were randomly divided into two groups ($n = 12$ per group) and administrated with free Scap-PBA or SCXN-PBA (50 mg kg$^{-1}$ Scap) by gavage. The jejunum (4 cm) of three mice in each group were harvested at certain time points (0.5, 1, 2, and 4 h). All the samples were fixed with 4% paraformaldehyde for 0.5 h. Then, the samples were sectioned, stained with 4′,6-diamidino-2-phenylindole, and photographed by a Nikon Eclipse C1 fluorescence microscope.

### Biodistribution
To explore the distribution of SCXN in vivo, Sxy-DiR was prepared by the same way as SCXN. After the CT26-luc mouse model was established, the mice were randomly divided into two groups ($n = 6$ per group) and orally administered with free DiR or Sxy-DiR (DiR concentration: 4 mg kg$^{-1}$). After gaseous anesthesia, all mice were injected intraperitoneally with luciferin (150 mg kg$^{-1}$) and photographed by IVIS (IVIS Spectrum, Perkin Elmer, USA) at certain time points (1, 4, 8, and 24 h). The fluorescence of DiR and the bioluminescence were both analyzed.

For more accurate display of the SCXN distribution in main tissues, the CT26 tumor-bearing mice were randomly divided into two groups ($n = 12$ per group), after orally administered with free DiR or Sxy-DiR (DiR concentration: 4 mg kg$^{-1}$), the major organs (heart, liver, spleen, lung, kidney, and whole intestine) and tumors of 3 mice in each group were taken out at fixed time points (1, 4, 8, and 24 h) and photographed by IVIS. Images were analyzed using IVIS Living Image Software (version 4.5.5).

For quantitative analysis, CT26 tumor-bearing mice were randomly divided into two groups ($n = 12$ per group) and orally administered with free Cap or SCXN (100 mg kg$^{-1}$ Cap). Then the major organs

(heart, liver, spleen, lung, and kidney) and tumors of 3 mice in each group were taken out at certain time points (1, 4, 8, and 24 h), homogenized, and centrifuged. The Cap concentrations in the supernatant were tested by HPLC. For the samples from tumors, the 5-Fu concentrations were also detected by HPLC.

## Pharmacokinetics

Mice were randomly divided into two groups ($n = 30$ per group) and administered orally with free Cap or SCXN (100 mg kg$^{-1}$ Cap). The blood of mice (500 μL) was taken into EP tubes equipped with 50 μL of heparin sodium (10 mg mL$^{-1}$) at 0 min, 15 min, 30 min, 1 h, 2 h, 4 h, 6 h, 8 h, 12 h, and 24 h, and then centrifuged (2000 g, 20 min). 200 μL of the supernatant was mixed with methanol (500 μL) and vortexed well. The mixture was centrifuged (2000 g, 20 min) again, finally the Cap concentration in the supernatant was tested by HPLC.

## Biocompatibility

Healthy female Balb/c mice were randomly assigned into 8 groups ($n = 3$ per group): saline, xylan, Sxy, Cap, Scap, xylan+Cap, Sxy+Scap, and SCXN. Mice were orally treated with various formulations (50 mg kg$^{-1}$ Cap, 500 mg kg$^{-1}$ xylan) every day for three weeks. During the evaluation, the body weights of mice were monitored every 3 days. At the end of 21 d, the whole blood of each group were harvested and the hematological and biochemical parameters were measured. Then the mice were sacrificed and main organs were sectioned and stained with H&E.

## Anti-tumor effect

CT26-luc tumor-bearing mice were randomly divided into 8 groups ($n = 9$ per group): saline, xylan, Sxy, Cap, Scap, xylan+Cap, Sxy+Scap, and SCXN. Mice were orally treated with various formulations (50 mg kg$^{-1}$ Cap, 500 mg kg$^{-1}$ xylan) every day for three weeks. The body weights were measured every 5 d, and the tumor growth was monitored according to the amounts of photons detected by IVIS after the intraperitoneal injection of luciferin (150 mg kg$^{-1}$). On the 21st day, three mice from each group were suffocated to death, and their tumors and intact intestines were taken out. The intestines were sectioned and stained with H&E. To evaluate CD8 and CD31 expression in tumor tissues, tumor sections were stained with primary antibody: anti-CD8 Rabbit Monoclonal Antibody (Servicebio, cat. no. GB13429, 1:800 dilution, https://shoposs.servicebio.cn/d/file/p/20200611/4a422e02568648e76b22abb6929e9e13.pdf) and anti-CD31 Mouse Monoclonal Antibody (Servicebio, cat. no. GB12063, 1:300 dilution, https://shoposs.servicebio.cn/2023/05/24/1684914947824vsNZzs.pdf), followed by staining with secondary antibody: Alexa488-conjugated Goat Anti-Rabbit IgG (Servicebio, cat. no. GB25303, 1:400 dilution, https://shoposs.servicebio.cn/d/file/p/20200508/2d79ecef85445afaad1a856b42a6db65.pdf) and CY3-conjugated Goat Anti-Mouse IgG (Servicebio, cat. no. GB21303, 1:300 dilution, https://shoposs.servicebio.cn/d/file/p/20200508/2d79ecef85445afaad1a856b42a6db65.pdf). To evaluate tumor cell apoptosis, TUNEL assay on tumor sections were carried out with Fluorescein (FITC) Tunel Cell Apoptosis Detection Kit (Servicebio). Sections were rinsed, stained with DAPI and observed with a Nikon Eclipse C1 microscope. All the antibodies were used following the suppliers' protocols. The remaining mice of each group were kept in the SPF grade environment until died. The evaluation ended at Day 45 and the survival curves and the median survivals were produced by GraphPad Prism software (version 8.3.0).

The therapy on MC38 tumor-bearing mice began when the tumor volumes reached approximately 100 mm$^3$. MC38 tumor-bearing mice were randomized into 8 groups ($n = 6$ per group): saline, xylan, Sxy, Cap, Scap, xylan+Cap, Sxy+Scap, and SCXN. Mice were orally treated with various formulations (50 mg kg$^{-1}$ Cap, 500 mg kg$^{-1}$ xylan) every day for three weeks. The body weights were measured every 3 d. The

tumor volumes of each group were measured and calculated ([major axis] × [minor axis]$^2$/2). At day 21, the mice of each group were asphyxiated to death. All the tumors were collected and photographed. Tumors were weighed and the tumor inhibiting rate (TIR) was calculated according to Eq. 3:

$$\text{TIR} = \left(1 - \frac{W\,test}{W\,saline}\right) \times 100\% \qquad (3)$$

$W_{test}$ and $W_{saline}$ in the equation meant the average tumor weight of the tested group and the saline group, respectively. The tumor sections were tested by TUNEL assays.

Maximal tumour size permitted by the ethics committee is 2000 mm$^3$. The maximal tumour size did not exceed this limit in this research.

## Anti-tumor immune response

The CT26 tumor-bearing mice were randomly divided into eight groups ($n = 3$ per group): saline, xylan, Sxy, Cap, Scap, xylan+Cap, Sxy+Scap, and SCXN. The formulations were administered orally (50 mg kg$^{-1}$ Cap, 500 mg kg$^{-1}$ xylan) every day. On the 21st day, mice were sacrificed, and the tumors and lymph nodes were taken out.

For DC maturation level evaluation, the lymph nodes were grinded in PBS (1 mL), filtered with nylon nets, and then centrifuged at 500 g for 15 min. After the supernatant was removed, 300 μL of PBS was added to resuspend cells. The samples were labeled with viability dye and blocked with Anti-mouse CD16/CD32 antibodies (TONBO Bioscience, cat. no. 70-0161-U100, Clone: 2.4G2, 1:100 dilution, https://cytekbio.com/products/purified-anti-mouse-cd16-cd32-2-4g2-fc-block). APC-eFluor 780-anti-mouse CD11c (eBioscience, cat. no. 12-0112-82, Clone: N418, 1:100 dilution, https://www.thermofisher.cn/cn/zh/antibody/product/CD11b-Antibody-clone-M1-70-Monoclonal/12-0112-82), PE-anti-mouse CD80 (eBioscience, cat. no. 12-0801-82, Clone: 16-10A1, 1:100 dilution, https://www.thermofisher.cn/cn/zh/antibody/product/CD80-B7-1-Antibody-clone-16-10A1-Monoclonal/12-0801-82), and APC-anti-mouse CD86 (eBioscience, cat. no. 17-0862-81, Clone: GL1, 1:100 dilution, https://www.thermofisher.cn/cn/zh/antibody/product/CD86-B7-2-Antibody-clone-GL1-Monoclonal/17-0862-81) were used to analyze DC maturation.

To investigate the tumor-infiltration of T lymphocytes, the tumor tissues were crushed, and treated with 2 mL of the mixture solution of 1 mg mL$^{-1}$ collagenase IV, 1 mg mL$^{-1}$ hyaluronidase, and 0.2 mg mL$^{-1}$ DNase I for 2 h at 37 °C. After removing the tumor debris with nylon nets, the filtrate was centrifuged at 500 g for 5 min, then the supernatant was removed. The cells were washed with PBS, stained with viability dye, and blocked with anti-mouse CD16/CD32 antibodies. The cells were then stained with the designated antibodies: APC-eFluor 780-CD45 Rat Monoclonal Antibody (eBioscience, cat. no. 47-0451-80, Clone: 30-F11, 1:100 dilution, https://www.thermofisher.cn/cn/zh/antibody/product/CD45-Antibody-clone-30-F11-Monoclonal/47-0451-80), PerCP-Cy5.5-anti-mouse CD3 (TONBO Bioscience, cat. no. 65-0031-U100, Clone: 145-2C11, 1:100 dilution, https://cytekbio.com/products/percp-cyanine5-5-anti-mouse-cd3e-145-2c11), PE-Cy7 anti-mouse CD4 (BioLegend, cat. 100527, Clone: RM4-5, 1:100 dilution, https://www.biolegend.com/en-us/products/pe-cyanine7-anti-mouse-cd4-antibody-1932), APC-anti-mouse CD8a (BioLegend, cat. no. 100711, Clone: 53-6.7, 1:100 dilution, https://www.biolegend.com/en-us/products/apc-anti-mouse-cd8a-antibody-150), PE-anti-mouse Foxp3 (TONBO Bioscience, cat. no. 50-5773-U025, Clone: 3G3, 1:100 dilution, https://cytekbio.com/products/pe-anti-mouse-foxp3-3g3).

All the antibodies were used following the suppliers' protocols. The expression levels of these markers were measured by flow cytometry (Fortessa, BD, USA). BD FACSDiva software (version 6.3.1) were used for data collection, and Flowjo software package (version 10.6.2)

was used for data analysis. Gating strategy is exemplified in Supplementary Figs. 9, 10.

### 16 S rDNA sequencing and SCFA measurement

The CT26 tumor-bearing mice were randomly divided into five groups ($n = 5$ per group): saline, xylan, Cap, Sxy+Scap, and SCXN. The mice were orally treated with the formulations (50 mg kg$^{-1}$ Cap, 500 mg kg$^{-1}$ xylan) every day for three weeks. On the Day 0 (before treatment), Day 10 (at the middle of treatment), and Day 20 (at the end of treatment) after the first administration, feces of each group were collected, quick-frozen in liquid nitrogen for 30 min, and stored at −80 °C. Microbiome DNA isolation and 16 S rDNA gene sequencing were completed with the help of Majorbio Co. Ltd., Shanghai. Isolation of microbial DNA from the feces of the mice was performed using a Qiagen E. Z. N. A.® Soil DNA Kit. The V3-V4 region of the 16 S rRNA-encoding gene was amplified from extracted DNA using the barcoded dual-index primers. The PCR product was identified by gel electrophoresis, purified by AxyPrep DNA Gel Extraction Kit and quantified by Quantus™ Fluorometer. The pooled amplicon library was then sequenced on the Illumina MiSeq platform using the NEXTFLEX Rapid DNA-Seq Kit according to the manufacturer's instructions. The raw data was processed on the Qiime2 (version 2022.2), including reducing sequencing and PCR errors, and denoising by DADA2 (filtering, dereplication, chimera identification, and merging paired-end reads and so on) to optimize the sequence and get amplicon sequence variants (ASVs) for taxonomic analysis. ASVs fewer than 0.1% in all the samples or annotated as chloroplast and mitochondrial contaminants were removed from all the samples. The sequencing numbers in all samples were normalized to minimal values. Based on the Silva 138/16 s bacteria database, species taxonomic analysis of ASVs was performed using the Naive bayes classifier. Alpha-diversity index was calculated by monther software (version 1.30). Different types of SCFA in feces were measured by GC-MS (TRACE 1310-ISQ LT, Thermo Fisher, USA). The intestine microbial abundance in healthy Balb/c mice was also analyzed.

### Statistics and reproducibility

All in vitro and in vivo experiments were carried out with at least 3 biological replicates for each experimental group. Biological replicates are indicated within figure legends. Data are expressed as the mean ± SD. Mice were assigned randomly to experimental groups. No statistical method was used to predetermine sample size. No samples were excluded from the analyses. Investigators were not blinded to allocation during experiments and outcome assessment, and the analyses were based on objectively measurable data. Unpaired two-sided Student's $t$-test were used when two groups were compared, with Welch's correction when variances are not equal. One-way ANOVA followed by Tukey's multiple comparisons post hoc test was used for testing differences between groups. A two-sided log-rank (Mantel−Cox) test was used for the statistical comparison of the survival study. All statistical analysis was conducted by the Prism software package (GraphPad Prism 8.3.0). Differences were considered to be statistically significant if $p < 0.05$.

### Reporting summary

Further information on research design is available in the Nature Portfolio Reporting Summary linked to this article.

## Data availability

All data generated or analysed during this study are included in this published article, Supplementary Information and Source data file. 16 s rDNA sequencing data have been deposited in the NCBI Sequence Read Archive; accession number: PRJNA986432). Species taxonomic analysis refer to Silva 138/16 s bacteria database (https://www.arb-silva.de/search/). H&E-stained tissue section images and flow cytometry data are available in the Science Data Bank; accession code: 31253.11.

sciencedb.09258. TEM images and LSCM images are available in the Science Data Bank; accession code: 31253.11.sciencedb.09096. Source data are provided with this paper.

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

## Acknowledgements

CT26 murine colon carcinoma cells and luciferase-expressing CT26 murine colon carcinoma (CT26-luciferase) cells were kindly provided by Prof. Yongzhuo Huang's lab in Shanghai Institute of Materia Medica, CAS. The authors are extremely grateful to Mass Spectrometry System at National Facility for Protein Science in Shanghai (NFPS), Zhangjiang Lab for instrument support and technical assistance. Engineer Mengmeng Wang from Cryo-Electron Microscopy Research Center in Shanghai Institute of Materia Medica, CAS is appreciated for her technical assistance on the characterization of the SCXN. National Key R&D Program of China (2022YFC3401404 to Y.-P.L.), National Natural Science Foundation of China (32171315 to Q.Y., 31930066 and 32130058 to Y.-P.L.), Natural Science Foundation of Shandong (ZR2019ZD25 to Y.-P.L.), and "Science and Technology Innovation Action Plan" Sailing

Plan of Shanghai (22YF1460500 to T.L.) are gratefully acknowledged for financial support.

## Author contributions

T.-Q.L., Q.Y. and Y.-P.L. proposed the project and designed the experiments. R.-Q.Z. and X.Z. synthesized the materials. T.-Q.L., R.-Q.Z., X.Z., W.T., and X.H. performed in vitro experiments. T.-Q.L., R.-Q.Z., X.Z., W.-L.Y., Y.L. and Y.-H.Z. conducted the in vivo experiments. T.-Q.L., R.-Q.Z. X.Z., and Q.Y. collected and analyzed the data. T.-Q.L., R.-Q.Z., Q.Y. and Y.-P.L. co-wrote the manuscript. All authors discussed the results and reviewed the manuscript.

## Competing interests

The authors declare no competing interests.
