## [Peer Review File · Nature Communications]

Combining gut microbiota modulation and chemotherapy by
capecitabine-loaded prebiotic nanoparticle improves
colorectal cancer therapyREVIEWER COMMENTS

Reviewer #1 (expertise: Prebiotics nanoparticles)- Remarks to the Author:

This manuscript is very interesting one because the authors aimed to treat colorectal cancer by combination therapy of gut microbiota modulation and chemotherapy using Cap-loaded prebiotics micelle. However, this paper cannot be accepted to this journal in this state. The comments are as followed.

Major comments

- 1) It is hard to be named as Sxy and Scap micelles because the particle sizes of micelles should be below 50 nm. They look like nanoparticles.
- 2) It should be explained why Sxy and Scap have the strong negative charges of zeta potentials because they look like complete reaction between stearic acid and xylan or stearic acid and Cap.
- 3) The hydrolysis of Scap in vitro and in vivo environments should be performed because it is totally different from drug activity of Cap and Scap.
- 4) It should be explained why the particle sizes of SCXM in page 5-111 fell to about 120 nm when incubated in AGT for 30 min.
- 5) What's difference between Cap and free Cap in page 5-117 ?
- 6) FITC-labelled Scap-loaded Sxy micelle should be used to measure the flow cytometer instead of coumarin-loaded Sxy micelle because Cap is totally different from coumarin in entering cells.
- 7) Cap concentration for the Scap-loaded SCXM should be checked in intra-tumoral drug accumulation instead of Scap because Scap looks like prodrug.
- 8) How can you check 5-Fu as the metabolite of Cap in page 8-174 ?

Minor comments

- 1) Please check the spelling of instine in Fig. 1-a-II.
- 2) Please check the size (d.nm) in Fig. 1-c,f.

Reviewer #2 (expertise: colorectal cancer research)- Remarks to the Author:

This manuscript details the generation of a bioengineered drug delivery system using xylan micelles, and demonstrates the use of these to deliver capecitabine (SCXM) into a murine model of colorectal cancer. There is an extensive amount of work detailed, in particular I find the study convincing in the following aspects:

- The rationale provided in the introduction provides a very clear series of reasons to undertake the study.
- The methods are clearly written, and provide sufficient detail for reproduction of some complex bioengineering and in vivo experimentation.
- Although not my primary area of expertise, the bioengineering data is clear, and the protocol for generating stable micelles appears to be reproducible.
- the animal data on drug responses is comprehensive - the time-course data presented in figure 2a-c is convincing showing retention rates of SCMx and demonstrates a significant amount of technically demanding work.
- the microbiome characterisation in figure 5 is well done and comprehensive, and demonstrates convincing changes to microbiota in response to SCXM.

However, I have some significant questions relating to the relationship between the drug delivery, immune response and microbiome elements of this study, and I have some major questions about these aspects and think this link should be made clearer before publication is warranted.

1. I am surprised to see in figure 3f that capecitabine alone has no effect on tumour growth, as there is clinical evidence of some effect albeit generally palliative (eg <https://www.ncbi.nlm.nih.gov/pmc/articles/PMC2386354/>). Does this data align with other studies on animal models? The authors mention the poor curative efficacy of systemic chemotherapy (line 71), the introduction would benefit from more balanced clinical information particularly on capecitabine efficacy to put this data into context as it is a key point to the study (line 72-79).

2. Data is presenting showing an impact of SCMx on cancer response (figure 2), immunity (figure 4) and microbiota (figure 5). But I am not convinced that these points necessarily relate together as the authors suggest, can the authors provide evidence that the altered microbiota mediates the drug response? Indeed in the discussion the authors acknowledge that xylan alone might be expected to alter SCFA profile (line 364)/microbiota balance (line 368), so I do not think the authors can claim that the chemotherapy response is "consequent" (line 362). A fecal transplant experiment may be convincing evidence of this if pre-administration of xylan doesn't work, but without experimental data to link these two observations this reads like two different studies.

3. In a similar vein, xylan alone does not alter immune responses as measured by CD8+ cells (figure 4h), which does not support the conclusions of the authors that SCMx alters microbiota, which then alters immune responses, which then alters drug responses. I would propose that this could be seen more as a series of correlations rather than causations, where immune and microbiota responses lie downstream of a more stable drug delivery system.

Minor comments:

- My only question on the characterisation data is on this would be what information the zeta potential provides (figure 1D)?
- Should figure 1c be frequency on the y axis, rather than intensity?
- could the C6 signal in figure 1h be amplified slightly, as I think the distribution looks nice but signal intensity was not easy to make out on my screen.
- can the authors improve the presentation of bar charts, the bars indicating statistical significance (eg figure 2d) are not easy to distinguish from the bars.
- There are some grammatical and language errors that would benefit from a proof-read. For example in introduction line 45, 56, 69 - authors should check throughout. I think it may also be more accurate to say that the prebiotic role of xylan has not been studied, rather than it is always neglected (line 66), and that it is a promising approach rather than "necessary to exploit" (line 67).
- Some of the results text (eg line 275-284) is more suitable for the discussion, where it could support the discussion of microbiome impacts of SCXM.
- Is there any way to summarise figure 3b/c? This type of data is not something I am familiar with, there are a large number of plots with no statistical analysis making it difficult to interpret the pattern of the data overall.

Reviewer #3 (expertise: cancer immunotherapy)- Remarks to the Author:

In this manuscript Lang and colleagues investigated the efficacy of capecitabine (Cap)-loaded micelles using a prebiotic (xylan-stearic acid [SCXM]) to treat experimental colorectal cancer (CRC). This is particularly suited for drug delivery to the lower gastrointestinal tract, as it can only be degraded by the microbes residing there. Furthermore, encapsulating Cap reduces its systemic side effects. They demonstrate that SCXM increases the half-life of Cap in blood and within the tumour. The claim that SCXM further potentiates anti-tumour immune responses, resulting in tumour growth retardation.

The immunomodulating capacity of chemotherapeutic regimens and prebiotics have received a considerable amount of interest in the recent decade and it is reassuring to see authors analysed the immune system components following treatment. However, I have major reservations regarding the flow cytometry data presented. Overall, the plots do not seem to have been set up adequately. For instance, there is a considerable overlap between CD4 and CD8 T cells in Figure 4d, which is surprising. Furthermore, double positive cells are seen here, which again may be due to voltage and compensation issues. In Figure 4a-b, it is essential to show the gating strategy for DCs, with representative plots for each treatment (perhaps as a supplemental figure). I have similar concerns for Figure 4g, where the subpopulations are not clearly separated and have a significant amount of overlap, making quantification impossible. Further, the quadrant gating is incorrect and should be adjusted on the Y axis. In Figure 4, the absolute number of all cell types indicated should be included. For instance, in Figure 4d, the percentage of CD8 cells is relatively consistent, whereas in Figure 4c, their absolute numbers are considerably higher. The same quantification should be done for DCs, CD4 T cells and Tregs. As such, I would not be able to trust

the interpretation of this part of the study and if the manuscript were considered to be accepted for publication, I recommend a full repeat of the experiments and data analysis. In summary, apart from the inadequate analysis of the immune landscape, this is a well designed study. However, it only investigated a single drug in a single tumour model (CT26), and fails to demonstrate broad applicability. Inclusions of patient-derived xenograft models and other complementary CRC models would help strengthen the findings.

Minor comments:

- Lines 56-57: Revise sentence.
- The authors state that the CRC model is 'orthotopic' but they inject the cells intrathoracically. An orthotopic model of CRC is whereby the tumour is engrafted via endoscopy, as described by Roper et al., Nature Biotech, 2017 (doi: 10.1038/nbt.3836.).
- The use of unconventional abbreviations such as CL and ITM should be avoided.
- The text size in Figure 5g should be increased.
- The bioavailability data in Figure 6 should be presented after the data shown in Figures 1-2. Additionally, higher magnification H&N images should be included in Figure 6b.
- Line 283: SCFA should be defined here.
- The manuscript may benefit from editing by a language expert (eg, mice model should be amended to mouse model).

Responses to reviewers

Reviewer #1

This manuscript is very interesting one because the authors aimed to treat colorectal cancer by combination therapy of gut microbiota modulation and chemotherapy using Cap-loaded prebiotics micelle. However, this paper cannot be accepted to this journal in this state. The comments are as followed.

Major comments

1) It is hard to be named as Sxy and Scap micelles because the particle sizes of micelles should be below 50 nm. They look like nanoparticles.

Response: The name “micelle” is changed to “nanoparticle” in the text.

2) It should be explained why Sxy and Scap have the strong negative charges of zeta potentials because they look like complete reaction between stearic acid and xylan or stearic acid and Cap.

Response: Xylan is a polysaccharide that is composed of (1,4)-linked β -D-xylopyranose units, which are partially substituted at C2 with 4-O-methyl- α -D-glucuronic acid units. This substituent group endows xylan with negative charges, which were displayed on the surface of SCXN. The discussion is supplemented at Page 5, Line 7-9. The diagram of the synthesis of Sxy is revised in Supplementary Figure 1.

3) The hydrolysis of Scap in vitro and in vivo environments should be performed because it is totally different from drug activity of Cap and Scap.

Response: The hydrolysis of Scap in vitro and in vivo are tested. The method and results are supplemented at Page 4, Line 29-30; Page 25, Line 6-18; Supplementary Figure 2.

4) It should be explained why the particle sizes of SCXM in page 5-111 fell to about 120 nm when incubated in AGT for 30 min.

Response: The possible reason is that in AGT with a low pH, the carboxyl groups in xylan were protonated and strong hydrogen bonds between the carboxyl groups and the hydroxyl groups formed, which led to the shrinkage of the nanoparticle. The explanation is supplemented in Page 5, Line 19-22.

5) What's difference between Cap and free Cap in page 5-117 ?

Response: The items “Cap” (actually should be Scap) in Page 5, Line 16 and Line 25 refer to the Scap in the Sxy/Scap nanoparticle. It's clarified in the text now (Page 5, Line 17 and 25).

6) FITC-labelled Scap-loaded Sxy micelle should be used to measure the flow cytometer instead of coumarin-loaded Sxy micelle because Cap is totally different from coumarin in entering cells.

Response: Since the conjugation of FITC with Scap is hard to achieve, we chose 1-

pyrenebutyric acid (PBA), which was similar to rhodamine, to label Scap by covalent conjugation. The results are similar to those of the C6-loaded Sxy nanoparticle and shown in Fig. 1h.

7) Cap concentration for the Scap-loaded SCXM should be checked in intratumoral drug accumulation instead of Scap because Scap looks like prodrug.

Response: Actually we did measure the concentration of cap in the biodistribution evaluation but not Scap. (Fig. 2d and 2e)

8) How can you check 5-Fu as the metabolite of Cap in page 8-174 ?

Response: The 5-Fu concentration in the tumor samples were tested by HPLC. The method is supplemented at Page 28, Line 3-4.

Minor comments

1) Please check the spelling of instine in Fig. 1-a-II.

Response: The word “intestine” is revised.

2) Please check the size (d.nm) in Fig. 1-c,f.

Response: The size unit is changed to “nm” to make it easier to understand.

Reviewer #2

This manuscript details the generation of a bioengineered drug delivery system using xylan micelles, and demonstrates the use of these to deliver capecitabine (SCXM) into a murine model of colorectal cancer. There is an extensive amount of work detailed, in particular I find the study convincing in the following aspects:

- The rationale provided in the introduction provides a very clear series of reasons to undertake the study.
- The methods are clearly written, and provide sufficient detail for reproduction of some complex bioengineering and in vivo experimentation.
- Although not my primary area of expertise, the bioengineering data is clear, and the protocol for generating stable micelles appears to be reproducible.
- the animal data on drug responses is comprehensive - the time-course data presented in figure 2a-c is convincing showing retention rates of SCM and demonstrates a significant amount of technically demanding work.
- the microbiome characterisation in figure 5 is well done and comprehensive, and demonstrates convincing changes to microbiota in response to SCXM.

However, I have some significant questions relating to the relationship between the drug delivery, immune response and microbiome elements of this study, and I have some major questions about these aspects and think this link should be made clearer before publication is warranted.

1. I am surprised to see in figure 3f that capecitabine alone has no effect on tumour growth, as there is clinical evidence of some effect albeit generally palliative (eg <https://www.ncbi.nlm.nih.gov/pmc/articles/PMC2386354/>). Does this data align with other studies on animal models? The authors mention the poor curative efficacy of systemic chemotherapy (line 71), the introduction would benefit from more balanced clinical information particularly on capecitabine efficacy to put this data into context as it is a key point to the study (line 72-79).

Response: One purpose of our study is to prolong the circulation time of Cap, so as to decrease the administration dose of Cap and reduce the side effects. In clinical applications, the dose is 2500 mg/m² per day for each patient, which is around 61 mg/kg. In our animal experiments, the dose is 50 mg/kg per day for each mouse, which is about 5.5 mg/kg per day when converted to the dose for a man. The dose of Cap in our study was only less than a tenth of that in clinical data. Therefore, the anti-tumor growth effect of free Cap without any carrier was very limited. The related explanation is supplemented at Page 19, Line 24-29.

2. Data is presenting showing an impact of SCM on cancer response (figure 2), immunity (figure 4) and microbiota (figure 5). But I am not convinced that these points necessarily relate together as the authors suggest, can the authors provide evidence that the altered microbiota mediates the drug response? Indeed in the discussion the authors acknowledge that xylan alone might be expected to alter

SCFA profile (line 364)/microbiota balance (line 368), so I do not think the authors can claim that the chemotherapy response is "consequent" (line 362). A fecal transplant experiment may be convincing evidence of this if pre-administration of xylan doesn't work, but without experimental data to link these two observations this reads like two different studies.

Response: Before our experiments were carried out, xylan was expected to alter gut microbiota and SCFA levels because it's a kind of well-known prebiotics, whose health-promoting functions have been validated and linked with its effect of increasing abundances or activity of probiotics (Carbohydrate Polymers 2021, 271, 118418). However, in our study, xylan alone hardly promoted probiotics abundances and SCFA production. The discrepancy between the results and the hypothesis may be due to the different microbial community composition in the colon cancer model from the normal gut (Data was supplemented in Fig. S11). The anti-tumor effect of xylan alone was poor, which was consistent with the results of gut microbiota analysis. Not only the xylan alone group, other experimental groups demonstrated a positive correlation between anti-tumor effects and increased probiotic abundance. We think the data indicated a promotion effect of SCXN, but not xylan alone, on tumor suppression. The explanation is supplemented at Page 19, Line 30-Page 20, Line 4; Page 20, Line 19-25.

3. In a similar vein, xylan alone does not alter immune responses as measured by CD8⁺ cells (figure 4h), which does not support the conclusions of the authors that SCMX alters microbiota, which then alters immune responses, which then alters drug responses. I would propose that this could be seen more as a series of correlations rather than causations, where immune and microbiota responses lie downstream of a more stable drug delivery system.

Response: It's explained in the manuscript that xylan alone had different effect from SCXM (now named SCXN) because the SCFA-producing probiotics which can degrade xylan had a much lower abundance in the intestines of colon tumor-bearing mice than that in the normal ones (See the response to question 2). The function of xylan could be enhanced when combined with Cap, which can be inferred from the results that the anti-tumor immune responses of xylan+Cap compared with xylan and that of Sxy+Scap versus Sxy were improved. The discussion is supplemented at Page 20, Line 25-Page 21, Line 2.

Minor comments:

- My only question on the characterisation data is on this would be what information the zeta potential provides (figure 1D)?

Response: As an essential index of nano particles, the zeta potential represents the surface charges of nano particles. Highly positive-charged drug delivery systems are prone to the clearance by the mononuclear-phagocyte system in vivo. Hence, it is necessary to provide the zeta potential data of SCXN. The explanation is supplemented at Page 5, Line 9-10.

- Should figure 1c be frequency on the y axis, rather than intensity?

Response: The particle sizes were measured by Zetasizer Nano-ZS90 (Malvern, UK). According to the original graph exported by the equipment (shown below), the y axis should be intensity (percent) rather than frequency.

- could the C6 signal in figure 1h be amplified slightly, as I think the distribution looks nice but signal intensity was not easy to make out on my screen.

Response: We re-run the experiment and replace C6 with a red dye, 1-pyrenebutyric acid (PBA)-labelled Scap. The new Fig. 1h is clearer than the old one.

- can the authors improve the presentation of bar charts, the bars indicating statistical significance (eg figure 2d) are not easy to distinguish from the bars.

Response: The statistical significances are shown with other symbols to make it easier to distinguish in Figure 2d now. The figure caption is also supplemented.

- There are some grammatical and language errors that would benefit from a proof-read. For example in introduction line 45, 56, 69 - authors should check throughout. I think it may also be more accurate to say that the prebiotic role of xylan has not been studied, rather than it is always neglected (line 66), and that it is a promising approach rather than "necessary to exploit" (line 67).

Response: We correct these errors and revise the sentences. (Page 3, Line 6, 18, 28-29, and 30)

- Some of the results text (eg line 275-284) is more suitable for the discussion, where it could support the discussion of microbiome impacts of SXCM.

Response: The description of gut microbiota functions is transferred from the results part (Page , Line) to the discussion part at Page 20, Line 4-15.

- Is there any way to summarise figure 3b/c? This type of data is not something I am familiar with, there are a large number of plots with no statistical analysis making it difficult to interpret the pattern of the data overall.

Response: The summarized data of Figure 3b and 3c (now are 4b and 4c) was shown in Supplementary Figure 4 and Supplementary Figure 5, respectively.

Reviewer #3

In this manuscript Lang and colleagues investigated the efficacy of capecitabine (Cap)-loaded micelles using a prebiotic (xylan-stearic acid [SCXM]) to treat experimental colorectal cancer (CRC). This is particularly suited for drug delivery to the lower gastrointestinal tract, as it can only be degraded by the microbes residing there. Furthermore, encapsulating Cap reduces its systemic side effects. They demonstrate that SCXM increases the half-life of Cap in blood and within the tumour. The claim that SCXM further potentiates anti-tumour immune responses, resulting in tumour growth retardation.

The immunomodulating capacity of chemotherapeutic regimens and prebiotics have received a considerable amount of interest in the recent decade and it is reassuring to see authors analysed the immune system components following treatment. However, I have major reservations regarding the flow cytometry data presented. Overall, the plots do not seem to have been set up adequately.

For instance, there is a considerable overlap between CD4 and CD8 T cells in Figure 4d, which is surprising. Furthermore, double positive cells are seen here, which again may be due to voltage and compensation issues.

Response: New flow cytometry images of intra-tumoral CD8⁺ T cells are shown in Figure 6d. The gating strategy is shown in Supplemental Fig. 10.

In Figure 4a-b, it is essential to show the gating strategy for DCs, with representative plots for each treatment (perhaps as a supplemental figure).

Response: The gating strategy for DCs is shown in Supplemental Fig. 9.

I have similar concerns for Figure 4g, where the subpopulations are not clearly separated and have a significant amount of overlap, making quantification impossible.

Response: New flow cytometry images of intra-tumoral Tregs are shown in Figure 6g. And the gating strategy for Tregs is shown in Supplemental Fig. 11.

Further, the quadrant gating is incorrect and should be adjusted on the Y axis.

Response: The gating of cells is based on the cluster of subpopulations but not the quadrant now.

In Figure 4, the absolute number of all cell types indicated should be included. For instance, in Figure 4d, the percentage of CD8 cells is relatively consistent, whereas in Figure 4c, their absolute numbers are considerably higher. The same quantification should be done for DCs, CD4 T cells and Tregs.

Response: The percentage of DCs and Tregs are replaced by their absolute numbers in Figure 6a and 6f.

As such, I would not be able to trust the interpretation of this part of the study and if the manuscript were considered to be accepted for publication, I recommend a full repeat of the experiments and data analysis.

Response: The immune responses analysis experiments in CT26 tumor-bearing mice were redone and new results are shown in the revised paper.

In summary, apart from the inadequate analysis of the immune landscape, this is a well designed study. However, it only investigated a single drug in a single tumour model (CT26), and fails to demonstrate broad applicability. Inclusions of patient-derived xenograft models and other complementary CRC models would help strengthen the findings.

Response: Another CRC model, the MC38 tumor mouse model, is built and the anti-tumor effect on this model is investigated. The results and methods are shown in Page 12, Line 22-29; Page 24, Line 1-2, 7-8, 15-16; Page 29, Line 6-17; Figure 5.

Minor comments:

- Lines 56-57: Revise sentence.

Response: This sentence is revised. (Page 3, Line 18-19)

- The authors state that the CRC model is ‘orthotopic’ but they inject the cells intrathoracically. An orthotopic model of CRC is whereby the tumour is engrafted via endoscopy, as described by Roper et al., Nature Biotech, 2017 (doi: 10.1038/nbt.3836.).

Response: We delete the word “orthotopic”. (Page 8, Line 8)

- The use of unconventional abbreviations such as CL and ITM should be avoided.

Response: CL and ITM are replaced with “plasma clearance” (Page 9, Line 8) and “immunosuppressive tumor microenvironment” (Page 15, Line 17), respectively.

- The text size in Figure 5g should be increased.

Response: The names of the species of microorganisms mentioned in this article is shown with larger font size in the new Figure 7g.

- The bioavailability data in Figure 6 should be presented after the data shown in Figures 1-2. Additionally, higher magnification H&N images should be included in Figure 6b.

Response: The biosafety evaluation data and Figure 6 (now is Figure 3) is presented after the data of pharmacokinetic study now (Page 10, Line 10). H&E images of hearts, liver, spleens, lungs, and kidneys are shown with higher magnification. For displaying intact structures of intestines, the magnification of intestine images is not changed.

- Line 283: SCFA should be defined here.

Response: The definition of SCFA is supplemented at Page 17, Line 20.

- The manuscript may benefit from editing by a language expert (eg, mice model should be amended to mouse model).

Response: The manuscript has been reviewed and edited. For example, revisions are at Page 2, Line 8; Page 3, Line 25; Page 4, Line 1, 13, 17; Page 5, Line 24; Page 8, Line 9; Page 10, Line 15, 18, 19; Page 14, Line 5; Page 17, Line 20; Page 18, Line 2; Page 27, Line 11; etc.

REVIEWER COMMENTS

Reviewer #1 (Remarks to the Author):

This manuscript can be accepted to this journal because the authors answered reviewer's comments.

Reviewer #2 (Remarks to the Author):

As in the review of the first submission, this is a study of combining chemotherapy with microbiota targeting for cancer therapy. The methodology is well explained, the rationale and need is well defined. Animal data is comprehensive and happy with the bioengineering.

The authors have addressed my concerns, there is additional data and additional explanation provided where necessary. I think there is a lot of work here, with generally good design.

Reviewer #3 (Remarks to the Author):

I am satisfied that the authors have now adequately addressed the majority of the points raised by the reviewers and I am glad to see the addition of MC38 tumour model data in Figure 5. However, I still have a number of reservations and suggestions, as detailed below.

Although the authors have improved the analyses and interpretation of the flow cytometry (FCM) data (eg, new plots in Figure 6d and 6g appear much more convincing), having reviewed the supplementary Figures 9-11, I still have major concerns about the gating strategies used throughout. Furthermore, to avoid any artefacts and background, all samples processed for FCM should have been initially quality checked by inclusion of a suitable live/dead marker and additionally doublets should have been excluded from the final analysis.

Overall, the represented leukocyte populations do not seem fully separated and in some cases are not visible (Sup Figure 9 -11). The current SSC-A vs CD marker FCM plots are not particularly a good way of defining such populations. Moreover, better mAb staining, live/dead staining and doublet cell exclusion along with more defined gating strategies are required to fully and confidently demonstrate the leukocyte subsets and allow accurate numeration. See examples here: <https://www.jci.org/articles/view/133353/figure/2>

Based on these, I strongly recommend that authors consider reviewing the data and repeat the staining using the standard procedures, as stated above.

Minor comments:

- Line 262: Figure 4, should state Supplementary Figure 3, not 2.
- Figure 5c: The same scale should be used on the Y axis in 5c. The values currently range from 500 to 2,000 and mask the differences at first glance.

Reviewer #4 (Remarks to the Author):

In this manuscript, the authors constructed a Cap-loaded nanoparticle using the prebiotics xylan-stearic acid conjugate (SCXN), which provides a promising CRC treatment by combining gut microbiota modulation and chemotherapy. The topic is interesting and the figures are of very high quality. This is a complex study and the proposed model is novel regarding the role of bacteria in chemotherapy resistance. Although all the data appear to support the model, there are a number of issues, particularly, concerning physiological relevance and data interpretation, make it questionable how much this really matters in humans.

1. Authors should employ stringent denoising and filtering of the 16S dataset to reduce the impact of potential contaminants and artifacts. 16S methods are prone to producing noise and using 16S data processing tools on default mode is not enough to produce clean datasets. The alpha and beta-diversity results also need to be supplemented. I strongly suggest the evaluation of this dataset using species-level taxonomic resolution offered by metagenomic sequencing which will lead to better biomarker identification.

2. The authors focus on changes of short-chain fatty acids in fecal metabolites, which is commonly accepted understanding. But the author failed to clarify how this change relate to gut bacteria and whether this is associated with the elevation of Akkermansia, Ruminococcus and Parabacteroides in the SCXN group. The potential mechanism by which SCXN causes changes in gut microbes should be elaborated more in detail. Since these two points address critical research gaps, the study team should provide more details with greater explanations to state the points clear.

3. The specific bacteria affected by chemotherapy should be analyzed, such as the relation to chemotherapy course, disease stage, prognosis and other indicators. Which bacteria genus or species affected by chemotherapy changed with the intervention of SCXN?

Response to reviewers

Reviewer #3

I am satisfied that the authors have now adequately addressed the majority of the points raised by the reviewers and I am glad to see the addition of MC38 tumour model data in Figure 5. However, I still have a number of reservations and suggestions, as detailed below.

Although the authors have improved the analyses and interpretation of the flow cytometry (FCM) data (eg, new plots in Figure 6d and 6g appear much more convincing), having reviewed the supplementary Figures 9-11, I still have major concerns about the gating strategies used throughout. Furthermore, to avoid any artefacts and background, all samples processed for FCM should have been initially quality checked by inclusion of a suitable live/dead marker and additionally doublets should have been excluded from the final analysis.

Overall, the represented leukocyte populations do not seem fully separated and in some cases are not visible (Sup Figure 9 -11). The current SSC-A vs CD marker FCM plots are not particularly a good way of defining such populations. Moreover, better mAb staining, live/dead staining and doublet cell exclusion along with more defined gating strategies are required to fully and confidently demonstrate the leukocyte subsets and allow accurate numeration. See examples here: <https://www.jci.org/articles/view/133353/figure/2>

Based on these, I strongly recommend that authors consider reviewing the data and repeat the staining using the standard procedures, as stated above.

Response: The flow cytometry analysis was redone with improved staining and gating strategy, including better mAb staining, live/dead staining and doublet cell exclusion. New results and gating strategies are shown in the revised manuscript (Page 15, Line 8-21; Fig. 6; Supplementary Fig. 9-10).

Minor comments:

1)- Line 262: Figure 4, should state Supplementary Figure 3, not 2.

Response: The caption of Figure 4 has been revised (Page 13, Line 5).

2)- Figure 5c: The same scale should be used on the Y axis in 5c. The values currently range from 500 to 2,000 and mask the differences at first glance.

Response: The Y axis of Figure 5c has been adjusted (Page 14).

Reviewer #4

In this manuscript, the authors constructed a Cap-loaded nanoparticle using the prebiotics xylan-stearic acid conjugate (SCXN), which provides a promising CRC treatment by combining gut microbiota modulation and chemotherapy. The topic is interesting and the figures are of very high quality. This is a complex study and the proposed model is novel regarding the role of bacteria in chemotherapy resistance. Although all the data appear to support the model, there are a number of issues, particularly, concerning physiological relevance and data interpretation, make it questionable how much this really matters in humans.

1. Authors should employ stringent denoising and filtering of the 16S dataset to reduce the impact of potential contaminants and artifacts. 16S methods are prone to producing noise and using 16S data processing tools on default mode is not enough to produce clean datasets.

Response: The fecal microbiota 16S analysis is redone with a mainstream microbiome analysis platform Qiime2 (Nat Biotechnol 2019, 37, 852-857), employing generally reliable denoising and filtering procedure to optimized sequencing result. The DADA2 used in Qiime2 (Nat Methods 2016, 13, 581-583) is an efficient denoising method, and it identified more real variants and output fewer spurious sequences than OTUs. Besides DADA2 denoising, we also removed low-level ASVs, chloroplast, and mitochondrial contaminants, and the sequencing numbers are normalized according to the minimal value to make the result more reliable. The data processing method in detail is supplemented (Page 32, Line 3-19). The new results are at Page 17, Line 9-18, 24-25.

The alpha and beta-diversity results also need to be supplemented.

Response: The alpha and beta-diversity analysis results are supplemented (Page 17, Line 5-9; Page 32, Line 19; Fig. 7a, b).

I strongly suggest the evaluation of this dataset using species-level taxonomic resolution offered by metagenomic sequencing which will lead to better biomarker identification.

Response: 16S rDNA analysis is a quick, simple, and inexpensive genomic analysis approach, which correlates well with genomic content and can be compared with large

existing public data sets (Nat Rev Microbiol 2018, 16, 410-422). It is widely applied in analyzing the change of microbiota (Adv Mater 2020, 32, 2004529; Nat Biomed Eng 2021, 5, 1377-1388; Nat Biomed Eng 2022, 6, 32-43). By contrast, metagenomic sequencing is relatively expensive, laborious, and complex, and contamination from host-derived DNA and organelles may obscure microbial signatures. Although metagenomic sequencing is a necessary tool to infer the relative abundance of microbial functional genes, in this study, which is focused on studying the microbial community composition, 16S analysis is enough to accomplish the aim.

2. The authors focus on changes of short-chain fatty acids in fecal metabolites, which is commonly accepted understanding. But the author failed to clarify how this change relate to gut bacteria and whether this is associated with the elevation of Akkermansia, Ruminococcus and Parabacteroides in the SCXN group. The potential mechanism by which SCXN causes changes in gut microbes should be elaborated more in detail. Since these two points address critical research gaps, the study team should provide more details with greater explanations to state the points clear.

Response: SCFAs in large intestines are mainly produced by gut microbiota when they metabolize indigestible oligosaccharide or polysaccharide, and currently no other major pathway has been reported. In this study, experimental animals were fed with the same food. Thus, the difference of gut SCFAs levels should be due to the distinct xylan intake. The elevated species including *Roseburia*, Ruminococcaceae, *Bifidobacterium*, and *Bacteroides* have been reported as main producers of SCFAs, especially butyric acid and propionic acid, which were restricted to specific bacteria. The explanation is supplemented at Page 20, Line 30-Page 21, Line 9. The potential mechanism by which SCXN causes changes in gut microbes is supplemented at Page 20, Line 6-19.

3. The specific bacteria affected by chemotherapy should be analyzed, such as the relation to chemotherapy course, disease stage, prognosis and other indicators. Which bacteria genus or species affected by chemotherapy changed with the intervention of SCXN?

Response: We re-designed the experiments and analyzed fecal microbiota of mice receiving different treatments at different disease stages and therapy courses. The methods are revised (Page 31, Line 25-Page 32, Line 1) and results are shown in Fig. 7

and Supplementary Fig. 11-14. Bacteria genus or species affected by chemotherapy changing with the intervention of SCXN are also analyzed and discussed (Page 17, Line 18-24; Page 20, Line 22-29).

REVIEWERS' COMMENTS

Reviewer #3 (Remarks to the Author):

The flow cytometry have been significantly improved and show the necessary QC. I am still not satisfied about CD4 vs Foxp3 staining in Figure 6g. My suggestion would be to first gate on CD4+ T cells and then show Foxp3+ cells. It is still evident that the CD4 antibody staining is not optimal.

Reviewer #5 (Remarks to the Author):

Thank you for your hard work in revising and extending this manuscript. Before publishing this work, I would strongly recommend another pass of copy-editing for language.

Response to reviewers

Reviewer #3

The flow cytometry have been significantly improved and show the necessary QC. I am still not satisfied about CD4 vs Foxp3 staining in Figure 6g. My suggestion would be to first gate on CD4+ T cells and then show Foxp3+ cells. It is still evident that the CD4 antibody staining is not optimal.

Response: The flow cytometry analysis was redone with improved gating strategy. New results and gating strategies are shown in the revised manuscript (Figure 6; Supplementary Figure 10).

Reviewer #5

Thank you for your hard work in revising and extending this manuscript. Before publishing this work, I would strongly recommend another pass of copy-editing for language.

Response: The English language editing services have been done in Editage (<http://app.editage.cn/>; The screenshot of part of the editing process is shown below). Now, the English language in the manuscript has been improved with clarity and readability. All the revised sections have been labeled in red color.